# AUTOMATED SELF-SUPERVISED LEARNING FOR GRAPHS

**Wei Jin** [*]
Michigan State University
jinwei2@msu.edu

**Xiaorui Liu**
Michigan State University
xiaorui@msu.edu

**Xiaoyu Zhao**
City University of Hong Kong
xy.zhao@cityu.edu.hk

**Yao Ma**
New Jersey Institute of Technology
yao.ma@njit.edu

**Neil Shah**
Snap Inc.
nshah@snap.com

**Jiliang Tang**
Michigan State University
tangjili@msu.edu

## ABSTRACT

Graph self-supervised learning has gained increasing attention due to its capacity to learn expressive node representations. Many pretext tasks, or loss functions have been designed from distinct perspectives. However, we observe that different pretext tasks affect downstream tasks differently across datasets, which suggests that searching over pretext tasks is crucial for graph self-supervised learning. Different from existing works focusing on designing single pretext tasks, this work aims to investigate how to automatically leverage multiple pretext tasks effectively. Nevertheless, evaluating representations derived from multiple pretext tasks without direct access to ground truth labels makes this problem challenging. To address this obstacle, we make use of a key principle of many real-world graphs, i.e., homophily, or the principle that "like attracts like," as the guidance to effectively search various self-supervised pretext tasks. We provide theoretical understanding and empirical evidence to justify the flexibility of homophily in this search task. Then we propose the AUTOSSL framework to automatically search over combinations of various self-supervised tasks. By evaluating the framework on 8 real-world datasets, our experimental results show that AUTOSSL can significantly boost the performance on downstream tasks including node clustering and node classification compared with training under individual tasks.

## 1 INTRODUCTION

Graphs are pivotal data structures describing the relationships between entities in various domains such as social media, biology, transportation and financial systems (Wu et al., 2019b; Battaglia et al., 2018). Due to their prevalence and rich descriptive capacity, pattern mining and discovery on graph data is a prominent research area with powerful implications. As the generalization of deep neural networks on graph data, graph neural networks (GNNs) have proved to be powerful in learning representations for graphs and associated entities (nodes, edges, subgraphs), and they have been employed in various applications such as node classification (Kipf & Welling, 2016a; Veličković et al., 2018), node clustering (Pan et al., 2018), recommender systems (Ying et al., 2018) and drug discovery (Duvenaud et al., 2015).

In recent years, the explosive interest in self-supervised learning (SSL) has suggested its great potential in empowering stronger neural networks in an unsupervised manner (Chen et al., 2020; Kolesnikov et al., 2019; Doersch et al., 2015). Many self-supervised methods have also been developed to facilitate graph representation learning (Jin et al., 2020; Xie et al., 2021; Wang et al., 2022) such as DGI (Veličković et al., 2019), PAR/CLU (You et al., 2020) and MVGRL (Hassani & Khasahmadi, 2020). Given graph and node attribute data, they construct pretext tasks, which are called SSL tasks, based on structural and attribute information to provide self-supervision for training graph neural networks without accessing any labeled data. For example, the pretext task of

---

[*] Work partially done while author was on internship at Snap Inc.

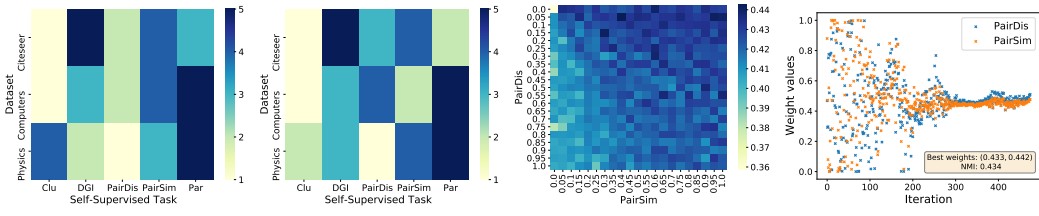

| (a) Node Clustering | (b) Node Classification | (c) Combining Two Tasks | (d) AUTOSSL |

Figure 1: **(a)(b):** Performance of 5 SSL tasks ranked best (1) to worst (5) by color on node clustering and classification, showing disparate performance across datasets and tasks. **(c):** Clustering performance heatmap on `Citeseer` when combining 2 SSL tasks, PAIRSIM and PAIRDIS, with different weights. **(d)** AUTOSSL's search trajectory for task weights, achieving near-ideal performance.

PAR is to predict the graph partitions of nodes. We examine how a variety of SSL tasks including DGI, PAR, CLU, PAIRDIS (Peng et al., 2020) and PAIRSIM (Jin et al., 2020; 2021) perform over 3 datasets. Their node clustering and node classification performance ranks are illustrated in Figure 1a and 1b, respectively. From these figures, we observe that different SSL tasks have distinct downstream performance cross datasets. This observation suggests that the success of SSL tasks strongly depends on the datasets and downstream tasks. Learning representations with a single task naturally leads to ignoring useful information from other tasks. As a result, searching SSL tasks is crucial, which motivates us to study on how to automatically compose a variety of graph self-supervised tasks to learn better node representations.

However, combining multiple different SSL tasks for unlabeled representation learning is immensely challenging. Although promising results have been achieved in multi-task self-supervised learning for computer vision, most of them assign equal weights to SSL tasks (Doersch & Zisserman, 2017; Ren & Lee, 2018; Zamir et al., 2018). Such combination might not always yield better performance than a single task, as different tasks have distinct importance according to specific dataset and downstream tasks. To illustrate this intuition, we combine two SSL tasks, PAIRDIS and PAIRSIM, with varied weights and illustrate the corresponding node clustering performance in Figure 1c. It clearly indicates that different choices of weights yield different performance. To circumvent this problem, we could plausibly search different weights for SSL tasks to optimize downstream tasks. However, to achieve such goal, we have two obstacles. First, the search space is huge, and thus search can be highly expensive. Hence, it is desirable to automatically learn these weights. Second, searching for optimal task weights typically requires guidance from downstream performance, which is naturally missing under the unsupervised setting. Thus, how to design an unsupervised surrogate evaluation measure that can guide the search process is necessary.

It is evident that many real-world graphs such as friendship networks, citation networks, co-authorship networks and co-purchase networks (McPherson et al., 2001; Shchur et al., 2018) satisfy the homophily assumption, i.e., "like attracts like", or that connected nodes tend to share the same label. This is useful prior knowledge, as it directly relates the label information of downstream tasks to the graph structure. In this work, we explicitly take advantage of this prior knowledge and assume that the predicted labels from good node embeddings should also adhere to homophily. Given the lack of ground-truth labels during SSL, we propose a pseudo-homophily measure to evaluate the quality of the node embeddings trained from specific combinations of SSL task. With pseudo-homophily, we are able to design an automated framework for SSL task search, namely AUTOSSL. Our work makes three significant contributions:

(1) To bridge the gap between unsupervised representation and downstream labels, we propose pseudo-homophily to measure the quality of the representation. Moreover, given graphs with high homophily, we theoretically show that pseudo-homophily maximization can help maximize the upper bound of mutual information between pseudo-labels and downstream labels.

(2) Based on pseudo-homophily, we propose two strategies to efficiently search SSL tasks, one employing evolution algorithm and the other performing differentiable search via meta-gradient descent. AUTOSSL is able to adjust the task weights during search as shown in Figure 1d.

(3) We evaluate the proposed AUTOSSL by composing various individual tasks on 8 real-world datasets. Extensive experiments have demonstrated that AUTOSSL can significantly improve

the performance of individual tasks on node clustering and node classification (e.g., up to *10.0%* relative improvement on node clustering).

## 2 BACKGROUND AND RELATED WORK

**Graph Neural Networks.** Graph neural networks (GNNs) are powerful tools for extracting useful information from graph data (Liu et al., 2021; Wu et al., 2019b; Kipf & Welling, 2016a; Veličković et al., 2018; Hamilton et al., 2017; Kipf & Welling, 2016b; Pan et al., 2018; Liu et al., 2020). They aim to learn a mapping function $f_\theta$ parameterized by $\theta$ to map the input graph into a low-dimensional space. Most graph neural networks follow a message passing scheme (Gilmer et al., 2017) where the node representation is obtained by aggregating the representation of its neighbors.

**Self-Supervised Learning in GNNs.** Graph neural networks have achieved superior performance in various applications; but they also require costly task-dependent labels to learn rich representations. To alleviate the need for the huge amount of labeled data, recent studies have employed self-supervised learning in graph neural networks to provide additional supervision (Jin et al., 2020; Veličković et al., 2019; You et al., 2020; Hassani & Khasahmadi, 2020; Hu et al., 2019; Qiu et al., 2020; Zhu et al., 2020b). Specifically, those SSL methods construct a pre-defined pretext task to assign pseudo-labels for unlabeled nodes/graphs and then train the model on the designed pretext task to learn representations. A recent work JOAO (You et al., 2021) on graph contrastive learning is proposed to automatically select data augmentation and focuses on graph classification task. Another related work is AUX-TS (Han et al., 2021), which also adaptively combines different SSL tasks but the combination happens at the fine-tuning stage and thus requires label information.

**Multi-Task Self-Supervised Learning.** Our work is also related to multi-task self-supervised learning (Doersch & Zisserman, 2017; Ren & Lee, 2018; Zamir et al., 2018). Most of them assume the tasks with equal weights and perform training under the supervised setting. But our work learns different weights for different tasks and does not require access to labeled data.

**Automated Loss Function Search.** Tremendous efforts have been paid to automate every aspect of machine learning applications (Yao et al., 2018; Liu et al., 2018; Zhao et al., 2021b), such as feature engineering, model architecture search and loss function search. Among them, our work is highly related to loss function search (Zhao et al., 2021a; Xu et al., 2018; Wang et al., 2020; Li et al., 2019). However, these methods are developed under the supervised setting and not applicable in self-supervised learning. Another related work, ELo (Piergiovanni et al., 2020), evolves multiple self-supervised losses based on Zipf distribution matching for action recognition. However, it is designed exclusively for image data and not applicable to non-grid graph-structured data. The problem of self-supervised loss search for graphs remains rarely explored. To bridge the gap, we propose an automated framework for searching SSL losses towards graph data in an unsupervised manner.

## 3 AUTOMATED SELF-SUPERVISED TASK SEARCH WITH AUTOSSL

In this section, we present the proposed framework of automated self-supervised task search, namely AUTOSSL. Given a graph $\mathcal{G}$, a GNN encoder $f_\theta(\cdot)$ and a set of $n$ self-supervised losses (tasks) $\{\ell_1, \ell_2, \ldots, \ell_n\}$, we aim at learning a set of loss weights $\{\lambda_1, \lambda_2, \ldots, \lambda_n\}$ such that $f_\theta(\cdot)$ trained with the weighted loss combination $\sum_{i=1}^{n} \lambda_i \ell_i$ can extract meaningful features from the given graph data. The key challenge is how to mathematically define "meaningful features". If we have the access to the labels of the downstream task, we can define "meaningful features" as the features (node embeddings) that can have high performance on the given downstream task. Then we can simply adopt the downstream performance as the optimization goal and formulate the problem of automated self-supervised task search as follows:

$$\min_{\lambda_1, \cdots, \lambda_n} \mathcal{H}(f_{\theta^*}(\mathcal{G})), \quad \text{s.t. } \theta^* = \arg\min_\theta \mathcal{L}(f_\theta, \{\lambda_i\}, \{\ell_i\}) = \arg\min_\theta \sum_{i=1}^{n} \lambda_i \ell_i(f_\theta(\mathcal{G})), \quad (1)$$

where $\mathcal{H}$ denotes the quality measure for the obtained node embeddings, and it can be any metric that evaluates the downstream performance such as cross-entropy loss for the node classification task. However, under the self-supervised setting, we do not have the access to labeled data and thus cannot employ the downstream performance to measure the embedding quality. Instead, we need an unsupervised quality measure $\mathcal{H}$ to evaluate the quality of obtained embeddings. In a nutshell, one challenge of automated self-supervised learning is: how to construct the goal of automated task search without the access to label information of the downstream tasks.

## 3.1 PSEUDO-HOMOPHILY

Most common graphs adhere to the principle of homophily, i.e., "birds of a feather flock together" (McPherson et al., 2001), which suggests that connected nodes often belong to the same class; e.g. connected publications in a citation network often have the same topic, and friends in social networks often share interests (Newman, 2018). Homophily is often calculated as the fraction of intra-class edges in a graph (Zhu et al., 2020a). Formally, it can be defined as follows,

**Definition 1** (Homophily). *The homophily of a graph $\mathcal{G}$ with node label vector $y$ is given by*

$$h(\mathcal{G}, y) = \frac{1}{|\mathcal{E}|} \sum_{(v_1, v_2) \in \mathcal{E}} \mathbb{1}(y_{v_1} = y_{v_2}), \tag{2}$$

*where $y_{v_i}$ indicates node $v_i$'s label and $\mathbb{1}(\cdot)$ is the indicator function.*

We calculate the homophily for seven widely used datasets as shown in Appendix A and we find that they all have high homophily, e.g., 0.93 in the `Physics` dataset. Considering the high homophily in those datasets, intuitively the predicted labels from the extracted node features should also have high homophily. Hence, the prior information of graph homophily in ground truth labels can serve as strong guidance for searching combinations of self-supervised tasks. As mentioned before, in self-supervised tasks, the ground truth labels are not available. Motivated by DeepCluster (Caron et al., 2018) which uses the cluster assignments of learned features as pseudo-labels to train the neural network, we propose to calculate the homophily based on the cluster assignments, which we term as *pseudo-homophily*. Specifically, we first perform $k$-means clustering on the obtained node embeddings to get $k$ clusters. Then the cluster results are used as the pseudo labels to calculate homophily based on Eq. (2). Note that though many graphs in the real world have high homophily, there also exist heterophily graphs (Zhu et al., 2020a; Pei et al., 2020) which have low homophily. We include a discussion on the homophily assumption in Appendix D.

**Theoretical analysis.** In this work, we propose to achieve self-supervised task search via maximizing pseudo-homophily. To understand its rationality, we develop the following theorem to show that pseudo-homophily maximization is related to the upper bound of mutual information between pseudo-labels and ground truth labels.

**Theorem 1.** *Suppose that we are given with a graph $\mathcal{G} = \{\mathcal{V}, \mathcal{E}\}$, a pseudo label vector $A \in \{0, 1\}^N$ and a ground truth label vector $B \in \{0, 1\}^N$ defined on the node set. We denote the homophily of $A$ and $B$ over $\mathcal{G}$ as $h_A$ and $h_B$, respectively. If the classes in $A$ and $B$ are balanced and $h_A < h_B$, the following results hold: (1) the mutual information between $A$ and $B$, i.e., MI(A,B), has an upper bound $\mathcal{U}_{A,B}$, where $\mathcal{U}_{A,B} = \frac{1}{N} \left[ 2\Delta \log\left(\frac{4}{N}\Delta\right) + 2\left(\frac{N}{2} - \Delta\right) \log\left(\frac{4}{N}\left(\frac{N}{2} - \Delta\right)\right) \right]$ with $\Delta = \frac{(h_B - h_A)|\mathcal{E}|}{2d_{max}}$ and $d_{max}$ denoting the largest node degree in the graph; (2) if $h_A < h_{A'} < h_B$, we have $\mathcal{U}_{A,B} < \mathcal{U}_{A',B}$.*

*Proof.* The detailed proof of this theorem can be found in Appendix B.

The above theorem suggests that a larger difference between pseudo-homophily and real homophily results in a lower upper bound of mutual information between the pseudo-labels and ground truth labels. Thus, maximizing pseudo-homophily is to maximize the upper bound of mutual information between pseudo-labels and ground truth labels, since we assume high homophily of the graph. Notably, while maximizing the upper bound does not guarantee the optimality of the mutual information, we empirically found that it works well in increasing the NMI value in different datasets, showing that it provides the right direction to promote the mutual information.

## 3.2 SEARCH ALGORITHMS

In the last subsection, we have demonstrated the importance of maximizing pseudo-homophily. Thus in the optimization problem of Eq. (1), we can simply set $\mathcal{H}$ to be negative pseudo-homophily. However, the evaluation of a specific task combination involves fitting a model and evaluating its pseudo-homophily, which can be highly expensive. Therefore, another challenge for automated self-supervised task search is how to design an efficient algorithm. In the following, we introduce the details of the search strategies designed in this work, i.e. AUTOSSL-ES and AUTOSSL-DS.

### 3.2.1 AUTOSSL-ES: EVOLUTIONARY STRATEGY

Evolution algorithms are often used in automated machine learning such as hyperparameter tuning due to their parallelism nature by design (Loshchilov & Hutter, 2016). In this work, we employ

the covariance matrix adaptation evolution strategy (CMA-ES) (Hansen et al., 2003), a state-of-the-art optimizer for continuous black-box functions, to evolve the combined self-supervised loss. We name this self-supervised task search approach as AUTOSSL-ES. In each iteration of CMA-ES, it samples a set of candidate solutions (i.e., task weights $\{\lambda_i\}$) from a multivariate normal distribution and trains the GNN encoder under the combined loss function. The embeddings from the trained encoder are then evaluated by $\mathcal{H}$. Based on $\mathcal{H}$, CMA-ES adjusts the normal distribution to give higher probabilities to good samples that can potentially produce a lower value of $\mathcal{H}$. Note that we constrain $\{\lambda_i\}$ in $[0, 1]$ and sample 8 candidate combinations for each iteration, which is trivially parallelizable as every candidate combination can be evaluated independently.

### 3.2.2 AUTOSSL-DS: DIFFERENTIABLE SEARCH VIA META-GRADIENT DESCENT

While the aforementioned AUTOSSL-ES is parallelizable, the search cost is still expensive because it requires to evaluate a large population of candidate combinations where every evaluation involves fitting the model in large training epochs. Thus, it is desired to develop gradient-based search methods to accelerate the search process. In this subsection, we introduce the other variant of our proposed framework, AUTOSSL-DS, which performs differentiable search via meta-gradient descent. However, pseudo-homophily is not differentiable as it is based on hard cluster assignments from $k$-means clustering. Next, we will first present how to make the clustering process differentiable and then introduce how to perform differentiable search.

**Soft Clustering.** Although $k$-means clustering assigns hard assignments of data samples to clusters, it can be viewed as a special case of Gaussian mixture model which makes soft assignments based on the posterior probabilities (Bishop, 2006). Given a Gaussian mixture model with centroids $\{\mathbf{c}_1, \mathbf{c}_2, \ldots, \mathbf{c}_k\}$ and fixed variances $\sigma^2$, we can calculate the posterior probability as follows:

$$p\left(\mathbf{x} \mid \mathbf{c}_i\right) = \frac{1}{\sqrt{2\pi\sigma^2}} \exp\left(-\frac{\|\mathbf{x} - \mathbf{c}_i\|_2}{2\sigma^2}\right), \tag{3}$$

where $\mathbf{x}$ is the feature vector of data samples. By Bayes rule and considering an equal prior, i.e., $p(\mathbf{c}_1) = p(\mathbf{c}_2) = \ldots = p(\mathbf{c}_k)$, we can compute the probability of a feature vector $\mathbf{x}$ belonging to a cluster $\mathbf{c}_i$ as:

$$p\left(\mathbf{c}_i \mid \mathbf{x}\right) = \frac{p\left(\mathbf{c}_i\right) p\left(\mathbf{x} \mid \mathbf{c}_i\right)}{\sum_j^k p\left(\mathbf{c}_j\right) p\left(\mathbf{x} \mid \mathbf{c}_j\right)} = \frac{\exp -\frac{(\mathbf{x} - \mathbf{c}_i)^2}{2\sigma^2}}{\sum_{j=1}^k \exp -\frac{(\mathbf{x} - \mathbf{c}_j)^2}{2\sigma^2}}. \tag{4}$$

If $\sigma \to 0$, we can obtain the hard assignments as the $k$-means algorithm. As we can see, the probability of each feature vector belonging to a cluster reduces to computing the distance between them. Then we can construct our homophily loss function as follows:

$$\mathcal{H}(f_{\theta^*}(\mathcal{G})) = \frac{1}{k|\mathcal{E}|} \sum_{i=1}^{k} \sum_{(v_1, v_2) \in \mathcal{E}} \ell\left(p(\mathbf{c}_i \mid \mathbf{x}_{v_1}), p(\mathbf{c}_i \mid \mathbf{x}_{v_2})\right), \tag{5}$$

where $\ell$ is a loss function measuring the difference between the inputs. With soft assignments, the gradient of $\mathcal{H}$ w.r.t. $\theta$ becomes tractable.

**Search via Meta Gradient Descent.** We now detail the differentiable search process for AUTOSSL-DS. A naive method to tackle bilevel problems is to alternatively optimize the inner and outer problems through gradient descent. However, we cannot perform gradient descent for the outer problem in Eq. (1) where $\mathcal{H}$ is not directly related to $\{\lambda_i\}$. To address this issue, we can utilize meta-gradients, i.e., gradients w.r.t. hyperparameters, which have been widely used in solving bi-level problems in meta learning (Finn et al., 2017; Zügner & Günnemann, 2019). To obtain meta-gradients, we need to backpropagate through the learning phase of the neural network. Concretely, the meta-gradient of $\mathcal{H}$ with respect to $\{\lambda_i\}$ is expressed as

$$\nabla_{\{\lambda_i\}}^{\text{meta}} := \nabla_{\{\lambda_i\}} \mathcal{H}(f_{\theta^*}(G)) \quad \text{s.t. } \theta^* = \text{opt}_\theta(\mathcal{L}(f_\theta, \{\lambda_i, \ell_i\})), \tag{6}$$

where $\text{opt}_\theta$ stands for the inner optimization that obtains $\theta^*$ and it is typically multiple steps of gradient descent. As an illustration, we consider $\text{opt}_\theta$ as $T + 1$ steps of vanilla gradient descent with learning rate $\epsilon$,

$$\theta_{t+1} = \theta_t - \epsilon \nabla_{\theta_t} \mathcal{L}(f_{\theta_t}, \{\lambda_i, \ell_i\}). \tag{7}$$

By unrolling the training procedure, we can express meta-gradient as

$$\nabla_{\{\lambda_i\}}^{\text{meta}} := \nabla_{\{\lambda_i\}} \mathcal{H}(f_{\theta_T}(G)) = \nabla_{f_{\theta_T}} \mathcal{H}(f_{\theta_T}(G)) \cdot [\nabla_{\{\lambda_i\}} f_{\theta_T}(G) + \nabla_{\theta_T} f_{\theta_T}(G) \nabla_{\{\lambda_i\}} \theta_T], \tag{8}$$

with $\nabla_{\{\lambda_i\}} \theta_T = \nabla_{\{\lambda_i\}} \theta_{T-1} - \epsilon \nabla_{\{\lambda_i\}} \nabla_{\theta_{T-1}} \mathcal{L}(f_{\theta_{T-1}}, \{\lambda_i, \ell_i\})$. Since $\nabla_{\{\lambda_i\}} f_{\theta_T}(G) = 0$, we have

$$\nabla_{\{\lambda_i\}}^{\text{meta}} := \nabla_{\{\lambda_i\}} \mathcal{H}(f_{\theta_T}(G)) = \nabla_{f_{\theta_T}} \mathcal{H}(f_{\theta_T}(G)) \cdot \nabla_{\theta_T} f_{\theta_T}(G) \nabla_{\{\lambda_i\}} \theta_T. \tag{9}$$

Note that $\theta_{T-1}$ also depends on the task weights $\{\lambda_i\}$ (see Eq. (7)). Thus, its derivative w.r.t. the task weights chains back until $\theta_0$. By unrolling all the inner optimization steps, we can obtain the meta-gradient $\nabla_{\{\lambda_i\}}^{\text{meta}}$ and use it to perform gradient descent on $\{\lambda_i\}$ to reduce $\mathcal{H}$:

$$\{\lambda_i\} \leftarrow \{\lambda_i\} - \eta \nabla_{\{\lambda_i\}}^{\text{meta}}, \tag{10}$$

where $\eta$ is the learning rate for meta-gradient descent (outer optimization).

However, if we use the whole training trajectory $\theta_0, \theta_1, \ldots, \theta_T$ to calculate the precise meta-gradient, it would have an extremely high memory footprint since we need to store $\theta_0, \theta_1, \ldots, \theta_T$ in memory. Thus, inspired by DARTS (Liu et al., 2018), we use an online updating rule where we only perform *one step gradient descent* on $\theta$ and then update $\{\lambda_i\}$ in each iteration. During the process, we constrain $\{\lambda_i\}$ in $[0, 1]$ and dynamically adjust the task weights in a differentiable manner. The detailed algorithm for AUTOSSL-DS is summarized in Appendix C.

## 4 EXPERIMENTAL EVALUATION

In this section, we empirically evaluate the effectiveness of the proposed AUTOSSL on self-supervised task search on real-world datasets. We aim to answer four questions as follows. **Q1:** Can AUTOSSL achieve better performance compared to training on individual SSL tasks? **Q2:** How does AUTOSSL compare to other unsupervised and supervised node representation learning baselines? **Q3:** Can we observe relations between AUTOSSL's pseudo-homophily objective and downstream classification performance? and **Q4:** How do the SSL task weights, pseudo-homophily objective, and downstream performance evolve during AUTOSSL's training?

### 4.1 EXPERIMENTAL SETTING

Since our goal is to enable automated combination search and discovery of SSL tasks, we use 5 such tasks including 1 contrastive learning method and 4 predictive methods – (1) **DGI** (Veličković et al., 2019): it is a contrastive learning method maximizing the mutual information between graph representation and node representation; (2) **CLU** (You et al., 2020), it predicts partition labels from Metis graph partition (Karypis & Kumar, 1998); (3) **PAR** (You et al., 2020), it predicts clustered labels from $k$-means clustering on node features; (4) **PAIRSIM** (Jin et al., 2020; 2021), it predicts pairwise feature similarity between node pairs and (5) **PAIRDIS** (Peng et al., 2020), it predicts shortest path length between node pairs. The proposed AUTOSSL framework automatically learns to jointly leverage the 5 above tasks and carefully mediate their influence. We also note that (1) the recent contrastive learning method, MVGRL (Hassani & Khasahmadi, 2020), needs to deal with a dense diffusion matrix and is prohibitively memory/time-consuming for larger graphs; thus, we only include it as a baseline to compare as shown in Table 2; and (2) the proposed framework is general and it is straightforward to combine other SSL tasks.

We perform experiments on 8 real-world datasets widely used in the literature (Yang et al., 2016; Shchur et al., 2018; Mernyei & Cangea, 2020; Hu et al., 2020), i.e., `Physics`, `CS`, `Photo`, `Computers`, `WikiCS`, `Citeseer`, `CoraFull`, and `ogbn-arxiv`. To demonstrate the effectiveness of the proposed framework, we follow (Hassani & Khasahmadi, 2020) and evaluate all methods on two different downstream tasks: node clustering and node classification. For the task of node clustering, we perform $k$-means clustering on the obtained embeddings. We set the number of clusters to the number of ground truth classes and report the normalized mutual information (NMI) between the cluster results and ground truth labels. Regarding the node classification task, we train a logistic regression model on the obtained node embeddings and report the classification accuracy on test nodes. *Note that labels are never used for self-supervised task search.* All experiments are performed under 5 different random seeds and results are averaged. Following DGI and MVGRL, we use a simple one-layer GCN (Kipf & Welling, 2016a) as our encoder and set the size of hidden dimensions to 512. We set $2\sigma^2 = 0.001$ and use L1 loss in the homophily loss function throughout the experiments. Further details of experimental setup can be found in Appendix A.

### 4.2 PERFORMANCE COMPARISON WITH INDIVIDUAL TASKS

To answer **Q1**, Table 1 summarizes the results for individual self-supervised tasks and AUTOSSL under the two downstream tasks, i.e., node clustering and node classification. From the table, we

Table 1: Performance comparison of self-supervised tasks (losses) on node clustering and node classification. The NMI rows indicate node clustering performance; ACC rows indicate node classification accuracy (%); P-H stands for pseudo-homophily. AUTOSSL regularly outperforms individual pretext tasks. (Bold: best in all methods; Underline: best in individual tasks). *Blue* and *red* numbers indicate the statistically significant improvements over the best individual task, via paired t-test at level 0.05 and 0.1, respectively (same for Table 2 and Table 3).

| Dataset | Metric | Self-Supervised Task | | | | | AUTOSSL | |
| --- | --- | --- | --- | --- | --- | --- | --- | --- |
| | | CLU | PAR | PAIRSIM | PAIRDIS | DGI | ES | DS |
| WikiCS | NMI | $0.177_{\pm 0.02}$ | $0.262_{\pm 0.02}$ | $\underline{0.341}_{\pm 0.01}$ | $0.169_{\pm 0.04}$ | $0.310_{\pm 0.02}$ | $\mathbf{0.366}_{\pm 0.01}$ | $0.344_{\pm 0.02}$ |
| | ACC | $74.19_{\pm 0.21}$ | $\underline{75.81}_{\pm 0.17}$ | $75.80_{\pm 0.17}$ | $75.28_{\pm 0.08}$ | $75.49_{\pm 0.17}$ | $\mathbf{76.80}_{\pm 0.13}$ | $76.58_{\pm 0.28}$ |
| | P-H | 0.549 | 0.567 | 0.693 | 0.463 | 0.690 | 0.751 | 0.749 |
| Citeseer | NMI | $0.318_{\pm 0.00}$ | $0.416_{\pm 0.00}$ | $0.428_{\pm 0.01}$ | $0.404_{\pm 0.01}$ | $\underline{0.439}_{\pm 0.00}$ | $\mathbf{0.449}_{\pm 0.01}$ | $\mathbf{0.449}_{\pm 0.01}$ |
| | ACC | $63.17_{\pm 0.19}$ | $69.25_{\pm 0.51}$ | $71.36_{\pm 0.42}$ | $70.72_{\pm 0.53}$ | $\underline{71.64}_{\pm 0.44}$ | $\mathbf{72.14}_{\pm 0.41}$ | $72.00_{\pm 0.32}$ |
| | P-H | 0.787 | 0.916 | 0.885 | 0.901 | 0.934 | 0.943 | 0.934 |
| Computers | NMI | $0.171_{\pm 0.00}$ | $\underline{0.433}_{\pm 0.00}$ | $0.387_{\pm 0.01}$ | $0.300_{\pm 0.01}$ | $0.318_{\pm 0.02}$ | $0.447_{\pm 0.01}$ | $\mathbf{0.448}_{\pm 0.01}$ |
| | ACC | $75.20_{\pm 0.20}$ | $\underline{87.26}_{\pm 0.15}$ | $82.64_{\pm 1.15}$ | $85.20_{\pm 0.41}$ | $83.45_{\pm 0.54}$ | $87.26_{\pm 0.64}$ | $\mathbf{88.18}_{\pm 0.43}$ |
| | P-H | 0.240 | 0.471 | 0.314 | 0.206 | 0.298 | 0.503 | 0.511 |
| CoraFull | NMI | $0.128_{\pm 0.00}$ | $\underline{0.498}_{\pm 0.00}$ | $0.409_{\pm 0.02}$ | $0.406_{\pm 0.01}$ | $0.462_{\pm 0.01}$ | $\mathbf{0.506}_{\pm 0.01}$ | $0.500_{\pm 0.00}$ |
| | ACC | $44.93_{\pm 0.53}$ | $57.54_{\pm 0.32}$ | $56.23_{\pm 0.59}$ | $58.48_{\pm 0.80}$ | $\underline{60.42}_{\pm 0.39}$ | $61.01_{\pm 0.50}$ | $\mathbf{61.10}_{\pm 0.68}$ |
| | P-H | 0.494 | 0.887 | 0.649 | 0.728 | 0.888 | 0.903 | 0.895 |
| CS | NMI | $0.658_{\pm 0.01}$ | $\underline{0.767}_{\pm 0.01}$ | $0.749_{\pm 0.01}$ | $0.635_{\pm 0.03}$ | $0.747_{\pm 0.01}$ | $\mathbf{0.772}_{\pm 0.01}$ | $0.771_{\pm 0.01}$ |
| | ACC | $88.58_{\pm 0.27}$ | $\underline{92.75}_{\pm 0.12}$ | $92.68_{\pm 0.09}$ | $89.56_{\pm 1.01}$ | $90.91_{\pm 0.51}$ | $93.26_{\pm 0.16}$ | $\mathbf{93.35}_{\pm 0.09}$ |
| | P-H | 0.845 | 0.882 | 0.886 | 0.786 | 0.883 | 0.895 | 0.890 |
| Physics | NMI | $0.692_{\pm 0.00}$ | $\underline{0.704}_{\pm 0.00}$ | $0.674_{\pm 0.00}$ | $0.420_{\pm 0.05}$ | $0.670_{\pm 0.00}$ | $0.725_{\pm 0.00}$ | $\mathbf{0.726}_{\pm 0.00}$ |
| | ACC | $93.60_{\pm 0.07}$ | $\underline{95.07}_{\pm 0.06}$ | $95.05_{\pm 0.10}$ | $91.69_{\pm 1.02}$ | $94.03_{\pm 0.15}$ | $\mathbf{95.57}_{\pm 0.02}$ | $95.13_{\pm 0.36}$ |
| | P-H | 0.911 | 0.913 | 0.905 | 0.821 | 0.906 | 0.921 | 0.923 |
| Photo | NMI | $0.327_{\pm 0.00}$ | $\underline{0.509}_{\pm 0.01}$ | $0.439_{\pm 0.04}$ | $0.293_{\pm 0.08}$ | $0.376_{\pm 0.03}$ | $\mathbf{0.560}_{\pm 0.04}$ | $0.515_{\pm 0.03}$ |
| | ACC | $90.33_{\pm 0.22}$ | $91.75_{\pm 0.25}$ | $91.13_{\pm 0.35}$ | $91.97_{\pm 0.32}$ | $\underline{92.08}_{\pm 0.37}$ | $92.04_{\pm 0.89}$ | $\mathbf{92.71}_{\pm 0.32}$ |
| | P-H | 0.434 | 0.602 | 0.428 | 0.327 | 0.401 | 0.791 | 0.616 |
| ogbn-arxiv | NMI | $0.305_{\pm 0.01}$ | $\underline{0.410}_{\pm 0.01}$ | $0.379_{\pm 0.01}$ | $0.314_{\pm 0.01}$ | $0.319_{\pm 0.01}$ | $0.424_{\pm 0.00}$ | $0.417_{\pm 0.00}$ |
| | ACC | $66.68_{\pm 0.34}$ | $67.90_{\pm 0.10}$ | $67.82_{\pm 0.20}$ | $67.63_{\pm 0.13}$ | $\underline{67.95}_{\pm 0.56}$ | $68.31_{\pm 0.05}$ | $\mathbf{69.13}_{\pm 0.04}$ |
| | P-H | 0.441 | 0.660 | 0.482 | 0.326 | 0.390 | 0.830 | 0.780 |
| Average Rank | NMI | 6.3 | 3.5 | 4.1 | 6.5 | 4.6 | 1.3 | 1.5 |
| | ACC | 6.8 | 3.9 | 4.9 | 5.4 | 3.9 | 1.8 | 1.4 |

make several observations. **Obs. 1:** individual self-supervised tasks have different node clustering and node classification performance for different datasets. For example, in Photo, DGI achieves the highest classification accuracy while PAR achieves the highest clustering performance; CLU performs better than PAIRDIS in both NMI and ACC on Physics while it cannot outperform PAIRDIS in WikiCS, Citeseer, Computers and CoraFull. This observation suggests the importance of searching suitable SSL tasks to benefit downstream tasks on different datasets. **Obs. 2:** Most of the time, combinations of SSL tasks searched by AUTOSSL can consistently improve the node clustering and classification performance over the best individual task on the all datasets. For example, the relative improvement over the best individual task on NMI from AUTOSSL-ES is *7.3%* for WikiCS and *10.0%* for Photo, and its relative improvement on ACC is *1.3%* for WikiCS. These results indicate that composing multiple SSL tasks can help the model encode different types of information and avoid overfitting to one single task. **Obs. 3:** We further note that individual tasks resulted in different pseudo-homophily as shown in the P-H rows of Table 1. Among them, CLU tends to result in a low pseudo-homophily and often performs much worse than other tasks in node clustering task, which supports our theoretical analysis in Section 3.1. It also demonstrates the necessity to increase pseudo-homophily as the two variants of AUTOSSL effectively search tasks that lead to higher pseudo-homophily. **Obs. 4:** The performance of AUTOSSL-ES and AUTOSSL-DS is close when their searched tasks lead to similar pseudo-homophily: the differences in pseudo-homophily, NMI and ACC are relative smaller in datasets other than Photo and Computers. It is worth noting that sometimes AUTOSSL-DS can even achieve higher pseudo-homophily than AUTOSSL-ES. This indicates that the online updating rule for $\{\lambda_i\}$ in AUTOSSL-DS not only can greatly reduce the searching time but also can generate good task combinations. In addition to efficiency, we highlight another major difference between them: AUTOSSL-ES directly finds the best task weights while AUTOSSL-DS adjusts the task weights to generate appropriate gradients to update model parameters. Hence, if we hope to find the best task weights and retrain the model, we should turn to AUTOSSL-ES. More details on their differences can be found in Appendix E.3.

### 4.3 PERFORMANCE COMPARISON WITH SUPERVISED AND UNSUPERVISED BASELINES

To answer **Q2**, we compare AUTOSSL with representative unsupervised and supervised node representation learning baselines. Specifically, for node classification we include 4 unsupervised baselines, i.e., GAE (Kipf & Welling, 2016b), VGAE (Kipf & Welling, 2016b), ARVGA (Pan et al., 2018) and MVGRL, and 2 supervised baselines, i.e. GCN and GAT (Veličković et al., 2018). We also provide the logistic regression performance on raw features and embeddings generated by a randomly initialized encoder, named as Raw-Feat and Random-Init, respectively. Note that *the two supervised baselines,* GCN *and* GAT*, use label information for node representation learning in an end-to-end manner, while other baselines and* AUTOSSL *do not leverage label information to learn representations.* The average performance and variances are reported in Table 2. From the table, we find that AUTOSSL outperforms unsupervised baselines in all datasets except `Citeseer` while the performance on `Citeseer` is still comparable to the state-of-the-art contrastive learning method MVGRL. When compared to supervised baselines, AUTOSSL-DS outperforms GCN and GAT in 4 out of 8 datasets, e.g., a *1.7%* relative improvement over GAT on `Computers`. AUTOSSL-ES also outperforms GCN and GAT in 3/4 out of 8 datasets. In other words, *our unsupervised representation learning* AUTOSSL *can achieve comparable performance with supervised representation learning baselines.* In addition, we use the same unsupervised baselines for node clustering and report the results in Table 3. Both AUTOSSL-ES and AUTOSSL-DS show highly competitive clustering performance. For instance, AUTOSSL-ES achieves *22.2%* and *27.5%* relative improvement over the second best baseline on `Physics` and `WikiCS`; AUTOSSL-DS also achieves *22.2%* and *19.8%* relative improvement on these two datasets. These results further validate that composing SSL tasks appropriately can produce expressive and generalizable representations.

Table 2: Node classification accuracy (%). The last two rows are supervised baselines. AUTOSSL consistently outperforms alternative self-supervised approaches, and frequently outperforms supervised ones. (Bold/Underline: best/runner-up among self-supervised approaches)

| Model | WikiCS | Citeseer | Computers | CoraFull | CS | Physics | Photo | ogbn-arxiv | Avg. Rank |
|---|---|---|---|---|---|---|---|---|---|
| Random-Init | $75.07_{\pm0.15}$ | $64.06_{\pm2.28}$ | $74.42_{\pm0.29}$ | $45.07_{\pm0.38}$ | $28.57_{\pm0.90}$ | $53.33_{\pm0.52}$ | $87.01_{\pm0.39}$ | $67.55_{\pm0.27}$ | 6.0 |
| Raw-Feat | $72.06_{\pm0.03}$ | $61.50_{\pm0.00}$ | $74.15_{\pm0.48}$ | $37.17_{\pm0.30}$ | $87.12_{\pm0.42}$ | $92.81_{\pm0.24}$ | $79.03_{\pm0.37}$ | $51.07_{\pm0.00}$ | 7.0 |
| GAE | $74.85_{\pm0.24}$ | $64.76_{\pm1.35}$ | $80.25_{\pm0.42}$ | $57.85_{\pm0.29}$ | $92.35_{\pm0.09}$ | $94.66_{\pm0.10}$ | $91.51_{\pm0.39}$ | $52.57_{\pm0.04}$ | 4.1 |
| VGAE | $74.16_{\pm0.16}$ | $67.50_{\pm0.42}$ | $81.05_{\pm0.41}$ | $53.72_{\pm0.30}$ | $92.15_{\pm0.16}$ | $94.58_{\pm0.17}$ | $88.98_{\pm1.05}$ | $52.00_{\pm0.19}$ | 4.6 |
| ARVGA | $71.64_{\pm1.03}$ | $46.88_{\pm2.15}$ | $67.61_{\pm0.92}$ | $45.20_{\pm1.33}$ | $87.26_{\pm1.07}$ | $93.84_{\pm0.13}$ | $77.74_{\pm1.16}$ | $31.57_{\pm2.96}$ | 7.1 |
| MVGRL | $75.89_{\pm0.12}$ | $\mathbf{72.36}_{\pm0.49}$ | $84.66_{\pm0.62}$ | $60.56_{\pm0.33}$ | $90.18_{\pm0.19}$ | $94.30_{\pm0.20}$ | $\underline{92.49}_{\pm0.40}$ | OOM | 3.1 |
| AUTOSSL-ES | $\mathbf{76.80}_{\pm0.13}$ | $\underline{72.14}_{\pm0.41}$ | $\underline{87.26}_{\pm0.64}$ | $\underline{61.01}_{\pm0.50}$ | $\underline{93.26}_{\pm0.16}$ | $\mathbf{95.57}_{\pm0.02}$ | $92.04_{\pm0.89}$ | $\underline{68.31}_{\pm0.05}$ | 1.9 |
| AUTOSSL-DS | $\underline{76.58}_{\pm0.28}$ | $72.00_{\pm0.32}$ | $\mathbf{88.18}_{\pm0.43}$ | $\mathbf{61.10}_{\pm0.68}$ | $\mathbf{93.35}_{\pm0.09}$ | $\underline{95.13}_{\pm0.36}$ | $\mathbf{92.71}_{\pm0.32}$ | $\mathbf{69.13}_{\pm0.04}$ | 1.5 |
| GCN | $76.42_{\pm0.02}$ | $71.26_{\pm0.15}$ | $87.53_{\pm0.21}$ | $63.77_{\pm0.37}$ | $93.04_{\pm0.09}$ | $95.66_{\pm0.15}$ | $93.09_{\pm0.11}$ | $71.74_{\pm0.29}$ | - |
| GAT | $77.30_{\pm0.01}$ | $71.00_{\pm0.62}$ | $86.74_{\pm0.69}$ | $63.73_{\pm0.43}$ | $92.53_{\pm0.19}$ | $95.54_{\pm0.08}$ | $92.30_{\pm0.28}$ | $71.46_{\pm0.34}$ | - |

Table 3: Clustering performance (NMI). AUTOSSL embeddings routinely exhibit superior NMI to alternatives. (Bold: best; Underline: runner-up).

| Model | WikiCS | Citeseer | Computers | CoraFull | CS | Physics | Photo | ogbn-arxiv | Avg. Rank |
|---|---|---|---|---|---|---|---|---|---|
| Random-Init | $0.107_{\pm0.02}$ | $0.354_{\pm0.03}$ | $0.155_{\pm0.01}$ | $0.318_{\pm0.01}$ | $0.716_{\pm0.02}$ | $0.551_{\pm0.01}$ | $0.246_{\pm0.04}$ | $0.306_{\pm0.01}$ | 6.4 |
| Raw-Feat | $0.182_{\pm0.00}$ | $0.316_{\pm0.00}$ | $0.166_{\pm0.00}$ | $0.215_{\pm0.00}$ | $0.642_{\pm0.00}$ | $0.489_{\pm0.00}$ | $0.282_{\pm0.00}$ | $0.150_{\pm0.01}$ | 7.1 |
| GAE | $0.243_{\pm0.02}$ | $0.313_{\pm0.02}$ | $0.441_{\pm0.00}$ | $0.485_{\pm0.00}$ | $0.731_{\pm0.01}$ | $0.545_{\pm0.06}$ | $\mathbf{0.616}_{\pm0.01}$ | $0.325_{\pm0.01}$ | 4.3 |
| VGAE | $0.261_{\pm0.01}$ | $0.364_{\pm0.01}$ | $0.423_{\pm0.00}$ | $0.453_{\pm0.01}$ | $0.733_{\pm0.01}$ | $0.563_{\pm0.02}$ | $0.530_{\pm0.04}$ | $0.311_{\pm0.01}$ | 4.0 |
| ARVGA | $0.287_{\pm0.02}$ | $0.191_{\pm0.02}$ | $0.237_{\pm0.01}$ | $0.301_{\pm0.01}$ | $0.616_{\pm0.03}$ | $0.526_{\pm0.05}$ | $0.301_{\pm0.01}$ | $0.201_{\pm0.01}$ | 6.4 |
| MVGRL | $0.263_{\pm0.01}$ | $\mathbf{0.452}_{\pm0.01}$ | $0.244_{\pm0.00}$ | $0.400_{\pm0.01}$ | $0.740_{\pm0.01}$ | $0.594_{\pm0.00}$ | $0.344_{\pm0.04}$ | OOM | 4.3 |
| AUTOSSL-ES | $\mathbf{0.366}_{\pm0.01}$ | $0.449_{\pm0.01}$ | $\underline{0.447}_{\pm0.01}$ | $\mathbf{0.506}_{\pm0.01}$ | $\mathbf{0.772}_{\pm0.01}$ | $\underline{0.725}_{\pm0.00}$ | $\underline{0.560}_{\pm0.04}$ | $\mathbf{0.424}_{\pm0.00}$ | 1.6 |
| AUTOSSL-DS | $\underline{0.344}_{\pm0.02}$ | $\underline{0.449}_{\pm0.01}$ | $\mathbf{0.448}_{\pm0.01}$ | $\underline{0.500}_{\pm0.00}$ | $\underline{0.771}_{\pm0.01}$ | $\mathbf{0.726}_{\pm0.00}$ | $0.515_{\pm0.03}$ | $\underline{0.417}_{\pm0.00}$ | 2.1 |

### 4.4 RELATION BETWEEN DOWNSTREAM PERFORMANCE AND PSEUDO-HOMOPHILY

In this subsection, we investigate the relation between downstream performance and pseudo-homophily and correspondingly answer **Q3**. Specifically, we use the candidate task weights sampled in the AUTOSSL-ES searching trajectory, and illustrate their node clustering (NMI) and node classification performance (ACC) with respect to their pseudo-homophily. The results on `Computers` and `WikiCS` are shown in Figure 2 and results for other datasets are shown in Appendix E.1. We observe that the downstream performance tends to be better if the learned embeddings tend to have higher pseudo-homophily. We also can observe that clustering performance has a clear relation with pseudo-homophily for all datasets. Hence, the results empirically support our theoretical analysis in Section 3.1 that lower pseudo-homophily leads to a lower upper bound of mutual information with ground truth labels. While classification accuracy has a less evident pattern, we can still observe that higher accuracy tends to concentrate on the high pseudo-homophily regions for 5 out of 7 datasets.

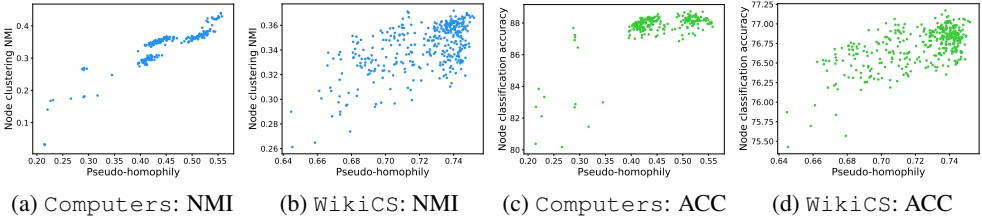

(a) `Computers`: NMI     (b) `WikiCS`: NMI     (c) `Computers`: ACC     (d) `WikiCS`: ACC

Figure 2: Relationship between downstream performance and pseudo-homophily.

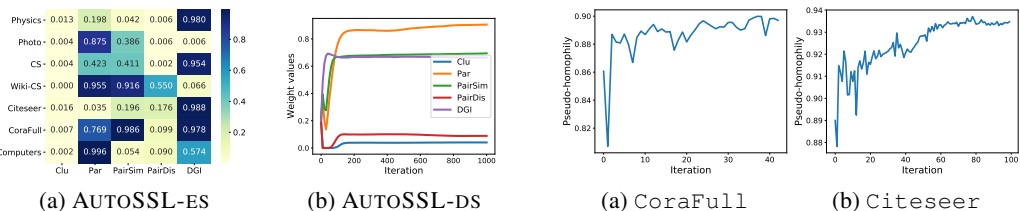

(a) AUTOSSL-ES      (b) AUTOSSL-DS        (a) `CoraFull`       (b) `Citeseer`

Figure 3: Visualization of Task Weights.      Figure 4: P-H change of AUTOSSL-ES

## 4.5 EVOLUTION OF SSL TASK WEIGHTS, PSEUDO-HOMOPHILY AND PERFORMANCE

To answer **Q4**, we visualize the final task weights searched by AUTOSSL-ES on all datasets through the heatmap in Figure 3a. From the figure, we make three observations. **Obs. 1:** The searched task weights vary significantly from dataset to dataset. For example, the weights for PAR and DGI are [0.198, 0.980] on `Physics` while they are [0.955, 0.066] on `WikiCS`. **Obs. 2:** In general, Par benefits co-purchase networks, i.e. `Photo` and `Computers`; DGI is crucial for citation/co-authorship networks, i.e. `Physics`, `CS`, `Citeseer`, and `CoraFull`. We conjecture that local structure information (the information that PAR captures) is essential for co-purchase networks while both local and global information (the information that DGI captures) are necessary in citation/co-authorship networks. **Obs. 3:** AUTOSSL-ES always gives very low weights to CLU, which indicates the pseudo-labels clustered from raw features are not good supervision on the selected datasets.

We also provide the evolution of task weights in AUTOSSL-DS for `CoraFull` dataset in Figure 3b. The weights of the 5 tasks eventually become stable: CLU and PAIRDIS are assigned with small values while PAIRSIM, DGI and CLU are assigned with large values. Thus, both AUTOSSL-ES and AUTOSSL-DS agree that PAIRDIS and PAR are less important for `CoraFull`.

We further investigate how pseudo-homophily changes over iterations. For AUTOSSL-ES, we illustrate the mean value of resulted pseudo-homophily in each iteration (round) in Figure 4. We only show the results on `CoraFull` and `Citeseer` while similar patterns are exhibited in other datasets. It is clear that AUTOSSL-ES can effectively increase the pseudo-homophily and thus search for better self-supervised task weights. The results for AUTOSSL-DS are deferred to Appendix E.2 due to the page limit.

## 5 CONCLUSION

Graph self-supervised learning has achieved great success in learning expressive node/graph representations. In this work, however, we find that SSL tasks designed for graphs perform differently on different datasets and downstream tasks. Thus, it is worth composing multiple SSL tasks to jointly encode multiple sources of information and produce more generalizable representations. However, without access to labeled data, it poses a great challenge in measuring the quality of the combinations of SSL tasks. To address this issue, we take advantage of graph homophily and propose pseudo-homophily to measure the quality of combinations of SSL tasks. We then theoretically show that maximizing pseudo-homophily can help maximize the upper bound of mutual information between the pseudo-labels and ground truth labels. Based on the pseudo-homophily measure, we develop two automated frameworks AUTOSSL-ES and AUTOSSL-DS to search the task weights efficiently. Extensive experiments have demonstrated that AUTOSSL is able to produce more generalize representations by combining various SSL tasks.

## ACKNOLWEDGEMENT

Wei Jin and Jiliang Tang are supported by the National Science Foundation (NSF) under grant numbers IIS1714741, CNS1815636, IIS1845081, IIS1907704, IIS1928278, IIS1955285, IOS2107215, and IOS2035472, the Army Research Office (ARO) under grant number W911NF-21-1-0198, the Home Depot, Cisco Systems Inc. and SNAP Inc.

## ETHICS STATEMENT

To the best of our knowledge, there are no ethical issues with this paper.

## REPRODUCIBILITY STATEMENT

To ensure reproducibility of our experiments, we provide our source code at `https://github.com/ChandlerBang/AutoSSL`. The hyper-parameters are described in details in the appendix. We also provide a pseudo-code implementation of our framework in the appendix.

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

## A  EXPERIMENTAL SETUP

**Dataset Statistics.** We evaluate the proposed framework on seven real-world datasets. The dataset statistics are shown in 4. All datasets can be loaded from PyTorch Geometric (Fey & Lenssen, 2019). When we evaluate the node classification performance, we need to use the training and test data. For `WikiCS` (Mernyei & Cangea, 2020), `ogbn-arxiv` (Hu et al., 2020) and `Citeseer` (Yang et al., 2016), we use the public data splits provided by the authors. For other datasets, we split the nodes into 10%/10%/80% for training/validation/test.

Table 4: Dataset statistics.

| Dataset | Network Type | #Nodes | #Edges | #Classes | #Features | Homophily |
|---|---|---|---|---|---|---|
| WikiCS | Reference network | 11,701 | 216,123 | 10 | 300 | 0.70 |
| CS | Co-authorship network | 18,333 | 81,894 | 15 | 6,805 | 0.81 |
| Physics | Co-authorship network | 34,493 | 247,962 | 5 | 8,415 | 0.93 |
| Computers | Co-purchase network | 13,381 | 245,778 | 10 | 767 | 0.78 |
| Photo | Co-purchase network | 7,487 | 119,043 | 8 | 745 | 0.83 |
| CoraFull | Citation network | 19,793 | 65,311 | 70 | 8,710 | 0.57 |
| Citeseer | Citation network | 3,327 | 4,732 | 6 | 3,703 | 0.74 |
| ogbn-arxiv | Citation network | 169,343 | 1,166,243 | 40 | 128 | 0.78 |

**Hyper-parameter Settings.** When calculating pseudo-homophily, we set the number of clusters to 10 for `ogbn-arxiv` and `Computers`, and 5 for other datasets. A small number of clusters can be more efficient but could be less stable. Following DGI (Veličković et al., 2019) and MVGRL (Hassani & Khasahmadi, 2020), we we use a simple one-layer GCN (Kipf & Welling, 2016a) as our encoder. We set the size of hidden dimensions to 512, weight decay to 0, dropout rate to 0. For individual SSL methods and AUTOSSL-ES, we set learning rate to 0.001, use Adam optimizer (Kingma & Ba, 2014), train the models with 1000 epochs and adopt early stopping strategy. For AUTOSSL-DS, we train the models with 1000 epochs and choose the model checkpoint that achieves the highest pseudo-homophily. We use Adam optimizer for both inner and outer optimization. The learning rate for outer optimization is set to 0.05. For AUTOSSL-ES, we use a population size of 8 for each round. Due to limited computational resources, we perform 80 rounds for `Citeseer`, 40 rounds for `CS`, `Computers`, `CoraFull`, `Photo`, `Physics`, `Computers` and `WikiCS`. We repeat the experiments on 5 different random seeds and report the mean values and variances for downstream performance. To fit DGI into GPU memory on larger datasets and accelerate its training, instead of using all the nodes we sample 2000 positive samples and 2000 negative samples for DGI on all datasets except `Citeseer`.

**Hardware and Software Configurations.** We perform experiments on one NVIDIA Tesla K80 GPU and one NVIDIA Tesla V100 GPU. Additionally, we use eight CPUs, with the model name as Intel(R) Xeon(R) Platinum 8260 CPU @ 2.40GHz. The operating system we use is CentOS Linux 7 (Core).

## B  PROOF

**Theorem 1.** *Suppose that we are given with a graph $\mathcal{G} = \{\mathcal{V}, \mathcal{E}\}$, a pseudo label vector $A \in \{0,1\}^N$ and a ground truth label vector $B \in \{0,1\}^N$ defined on the node set. We denote the homophily of $A$ and $B$ over $\mathcal{G}$ as $h_A$ and $h_B$, respectively. If the classes in $A$ and $B$ are balanced and $h_A ¡ h_B$, the following results hold: (1) the mutual information between $A$ and $B$, i.e., MI(A,B), has an upper bound $\mathcal{U}_{A,B}$, where $\mathcal{U}_{A,B} = \frac{1}{N}\left[2\Delta \log(\frac{4}{N}\Delta) + 2(\frac{N}{2} - \Delta)\log\left(\frac{4}{N}(\frac{N}{2} - \Delta)\right)\right]$ with $\Delta = \frac{(h_B - h_A)|\mathcal{E}|}{2d_{max}}$ and $d_{max}$ denoting the largest node degree in the graph; (2) if $h_A < h_{A'} < h_B$, we have $\mathcal{U}_{A,B} < \mathcal{U}_{A',B}$.*

**Proof.** (1) We start with the proof of the first result. The mutual information between two random variables $X$ and $Y$ is expressed as

$$MI(X,Y) = \sum_{y \in \mathcal{Y}} \sum_{x \in \mathcal{X}} p_{(A,B)}(x,y) \log\left(\frac{p_{(X,Y)}(x,y)}{p_X(x)p_Y(y)}\right). \tag{11}$$

Let $\mathcal{A}_i$ and $\mathcal{B}_i$ denote the set of nodes in the $i$-th class in $A$ and $B$, respectively. Following the definition in Eq. (11), the mutual information between $A$ and $B$ can be formulated as,

$$MI(A, B) = \sum_{i=0}^{n_A-1} \sum_{j=0}^{n_B-1} \frac{|\mathcal{A}_i \cap \mathcal{B}_j|}{N} \log \frac{N |\mathcal{A}_i \cap \mathcal{B}_j|}{|\mathcal{A}_i| |\mathcal{B}_j|}, \tag{12}$$

where $n_A, n_B$ denote the number of classes in $A$ and $B$. Since here we only consider 2 classes in $A$ and $B$, we have

$$MI(A, B) = \frac{|\mathcal{A}_0 \cap \mathcal{B}_0|}{N} \log \frac{N |\mathcal{A}_0 \cap \mathcal{B}_0|}{|\mathcal{A}_0| |\mathcal{B}_0|} + \frac{|\mathcal{A}_0 \cap \mathcal{B}_1|}{N} \log \frac{N |\mathcal{A}_0 \cap \mathcal{B}_1|}{|\mathcal{A}_0| |\mathcal{B}_1|}$$
$$+ \frac{|\mathcal{A}_1 \cap \mathcal{B}_0|}{N} \log \frac{N |\mathcal{A}_1 \cap \mathcal{B}_0|}{|\mathcal{A}_1| |\mathcal{B}_0|} + \frac{|\mathcal{A}_1 \cap \mathcal{B}_1|}{N} \log \frac{N |\mathcal{A}_1 \cap \mathcal{B}_1|}{|\mathcal{A}_1| |\mathcal{B}_1|}. \tag{13}$$

Let $|\mathcal{A}_0 \cap \mathcal{B}_0| = x$, $|\mathcal{A}_0| = a$ and $|\mathcal{B}_0| = b$. We then have

$$\begin{cases} |\mathcal{A}_0| + |\mathcal{A}_1| = N \Rightarrow |\mathcal{A}_1| = N - a, \\ |\mathcal{B}_0| + |\mathcal{B}_1| = N \Rightarrow |\mathcal{B}_1| = N - b, \\ |\mathcal{A}_0 \cap \mathcal{B}_0| + |\mathcal{A}_0 \cap \mathcal{B}_1| = |\mathcal{A}_0| \Rightarrow |\mathcal{A}_0 \cap \mathcal{B}_1| = a - x, \\ |\mathcal{A}_0 \cap \mathcal{B}_0| + |\mathcal{A}_1 \cap \mathcal{B}_0| = |\mathcal{B}_0| \Rightarrow |\mathcal{A}_1 \cap \mathcal{B}_0| = b - x, \\ |\mathcal{A}_0 \cap \mathcal{B}_1| + |\mathcal{A}_1 \cap \mathcal{B}_1| = |\mathcal{B}_1| \Rightarrow |\mathcal{A}_1 \cap \mathcal{B}_1| = N - b - a + x. \end{cases} \tag{14}$$

With the equations above, we rewrite $MI(A, B)$ as follows,

$$MI(A, B) = \frac{1}{N} [x \log \frac{Nx}{ab} + (a - x) \log \frac{N(a - x)}{a(N - b)}$$
$$+ (b - x) \log \frac{N(b - x)}{(N - a)b} + (N - b - a + x) \log \frac{N(N - b - a + x)}{(N - a)(N - b)}]. \tag{15}$$

Then we rewrite result (1) in the theorem as an optimization problem,

$$\max f(x) = MI(A, B) \tag{16}$$

with constraints,

$$\begin{cases} 0 \le |\mathcal{A}_0 \cap \mathcal{B}_0| \le |\mathcal{A}_0| \Rightarrow 0 \le x \le a, \\ 0 \le |\mathcal{A}_0 \cap \mathcal{B}_0| \le |\mathcal{B}_0| \Rightarrow 0 \le x \le b, \\ |\mathcal{A}_1 \cap \mathcal{B}_1| \ge 0 \Rightarrow x \ge a + b - N, \\ |\mathcal{A}_0 \cap \mathcal{B}_0| + |\mathcal{A}_1 \cap \mathcal{B}_1| \le N \Rightarrow x \le \frac{a+b}{2}, \\ |\mathcal{A}_0 \cap \mathcal{B}_1| + |\mathcal{A}_1 \cap \mathcal{B}_0| \le N \Rightarrow x \ge \frac{a+b-N}{2}, \end{cases} \tag{17}$$

Note that the equality of $|\mathcal{A}_0 \cap \mathcal{B}_0| + |\mathcal{A}_1 \cap \mathcal{B}_1| \le N$ holds when $A$ and $B$ are the same. However, $A$ and $B$ have different homophily, which indicates $|\mathcal{A}_0 \cap \mathcal{B}_0| + |\mathcal{A}_1 \cap \mathcal{B}_1|$ cannot reach $N$ (the same for $|\mathcal{A}_0 \cap \mathcal{B}_1| + |\mathcal{A}_1 \cap \mathcal{B}_0|$). Let $\mathcal{E}_A, \mathcal{E}_B$ denote the inter-class edges for $A$ and $B$, respectively. Thus, $h_A = 1 - \frac{|\mathcal{E}_A|}{|\mathcal{E}|}$ and $h_B = 1 - \frac{|\mathcal{E}_B|}{|\mathcal{E}|}$. Since $h_A < h_B$, we have $|\mathcal{E}_A| > |\mathcal{E}_B|$. This indicates that there are at least $|\mathcal{E}_A| - |\mathcal{E}_B|$ edges in $A$ connecting nodes that belong to the same ground truth class, as shown in Figure 5.

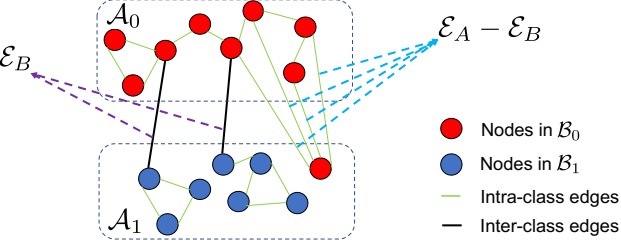

Figure 5: Illustration for $|\mathcal{A}_0 \cap \mathcal{B}_0| + |\mathcal{A}_1 \cap \mathcal{B}_1|$. The two dashed rectangles divide the nodes into $\mathcal{A}_0$ and $\mathcal{A}_1$; red and blue nodes denote nodes in $\mathcal{B}_0$ and $\mathcal{B}_1$, respectively.

Let $d_{\max}$ denote the maximum degree in the graph and we know that at least $\frac{|\mathcal{E}_A| - |\mathcal{E}_B|}{d_{\max}}$ nodes are "misplaced" in $A$, e.g., in Figure 5 the red node in $\mathcal{A}_1$ should be placed in $\mathcal{A}_0$ to achieve $|\mathcal{A}_0 \cap \mathcal{B}_0| +$

$|\mathcal{A}_1 \cap \mathcal{B}_1| = N$. Let $\Delta = \frac{|\mathcal{E}_A| - |\mathcal{E}_B|}{2d_{\max}} = \frac{(h_B - h_A)|\mathcal{E}|}{2d_{\max}}$, and we have $|\mathcal{A}_0 \cap \mathcal{B}_0| + |\mathcal{A}_1 \cap \mathcal{B}_1| \le N - 2\Delta$ and $|\mathcal{A}_0 \cap \mathcal{B}_1| + |\mathcal{A}_1 \cap \mathcal{B}_0| \le N - 2\Delta$.

With the new constraints, we rewrite the optimization problem as

$$\max f(x) = MI(A, B) \quad \text{s.t.} \begin{cases} 0 \le x \le a, \\ x \le b, \\ x \ge a + b - N, \\ x \le \frac{a+b-2\Delta}{2}, \\ x \ge \frac{a+b-(N-2\Delta)}{2}, \end{cases} \tag{18}$$

Further, the derivative of $f(x)$ is expressed as follows,

$$f'(x) = \frac{1}{N} \log \frac{x(N - b - a + x)}{(a - x)(b - x)}. \tag{19}$$

Let $f'(x) > 0$, we have $x > \frac{ab}{N}$; let $f'(x) < 0$, we have $x < \frac{ab}{N}$. Thus, $f(x)$ is monotonically decreasing at $[\max(0, a + b - N, \frac{a+b-(N-2\Delta)}{2}), \frac{ab}{N}]$ and monotonically increasing $[\frac{ab}{N}, \min(a, b, \frac{a+b-2\Delta}{2})]$.

Note that in the theorem we assume the pseudo-labels and ground truth classes are balanced, i.e., $a = b = \frac{N}{2}$. Then $MI(A, B)$ becomes,

$$MI(A, B) = f(x) = \frac{1}{N} \left[ 2x \log \frac{4}{N} x + 2(\frac{N}{2} - x) \log \left( \frac{4}{N}(\frac{N}{2} - x) \right) \right]. \tag{20}$$

Hence, $f(x)$ is monotonically decreasing at $[\Delta, \frac{N}{4}]$ and monotonically increasing $[\frac{N}{4}, \frac{N}{2} - \Delta]$. So the maximum value of $f(x)$ is at either $x = \Delta$ or $x = \frac{N}{2} - \Delta$. Further it is easy to know that $f(\frac{N}{4} - x) = f(\frac{N}{4} + x)$. Then we have $f(\Delta) = f(\frac{N}{2} - \Delta)$, and we can get the maximum value of $f(x)$ as follows,

$$\mathcal{U}_{A,B} = \max f(x) = f(\Delta) = \frac{1}{N} \left[ 2\Delta \log(\frac{4}{N} \Delta) + 2(\frac{N}{2} - \Delta) \log \left( \frac{4}{N}(\frac{N}{2} - \Delta) \right) \right] \tag{21}$$

with $\Delta = \frac{(h_B - h_A)|\mathcal{E}|}{2d_{\max}}$. In other words, $MI(A, B)$ reaches its upper bound when $|\mathcal{A}_0 \cap \mathcal{B}_0| = \frac{(h_B - h_A)|\mathcal{E}|}{2d_{\max}}$ or $\frac{N}{2} - \frac{(h_B - h_A)|\mathcal{E}|}{2d_{\max}}$.

(2) From the constraints $x \le \frac{a+b-2\Delta}{2}$ and $x \ge \frac{a+b-(N-2\Delta)}{2}$ in Eq (18), we have $\frac{a+b-(N-2\Delta)}{2} \le \frac{a+b-2\Delta}{2} \Rightarrow \Delta \le \frac{N}{4}$. Based on the discussion in (1), we know that $f(\Delta)$ is monotonically decreasing at $[0, \frac{N}{4}]$, which means an increase of $\Delta$ leads to a decrease in $f(\Delta)$, i.e., a smaller value of $\mathcal{U}_{A,B}$. Since $\Delta = \frac{(h_B - h_A)|\mathcal{E}|}{2d_{\max}}$, a decrease in $h_A$ will lead to a increase in $\Delta$. Then we have $\mathcal{U}_{A,B} < \mathcal{U}_{A',B}$ if $h_A < h_{A'} < h_B$.

**Remark on a more generalized case.** We now discuss the case where we do not have assumptions on $a$ and $b$. As we demonstrated in the above discussion, $f(x)$ is monotonically decreasing at $[\max(0, a + b - N, \frac{a+b-(N-2\Delta)}{2}), \frac{ab}{N}]$ and monotonically increasing $[\frac{ab}{N}, \min(a, b, \frac{a+b-2\Delta}{2})]$. Thus, the maximum value of $f(x)$ should be one of the values of $f(0), f(a + b - N), f(\frac{a+b-(N-2\Delta)}{2}), f(a), f(b)$ and $f(\frac{a+b-2\Delta}{2})$. As our goal is to show that $\mathcal{U}_{A,B}$ would be small with low $h_A$, to simplify the analysis, we consider a large value of $\Delta$ (or a small value of $h_A$) which satisfies $\Delta \ge \frac{1}{2}|N - (a + b)|$ and $\Delta \ge \frac{1}{2}|a - b|$. This indicates $x$ is bounded by $[\frac{a+b-(N-2\Delta)}{2}, \frac{a+b-2\Delta}{2}]$. Then the maximum value of $f(x)$, i.e., $\mathcal{U}_{A,B}$, is expressed as

$$\mathcal{U}_{A,B} = \max(f(\frac{a + b - (N - 2\Delta)}{2}), f(\frac{a + b - 2\Delta}{2})). \tag{22}$$

When $\frac{a+b-(N-2\Delta)}{2} \le \frac{ab}{N} \le \frac{a+b-2\Delta}{2}$, it is easy to see that larger $\Delta$ (or smaller $h_A$) will lead to smaller $\mathcal{U}_{A,B}$ because both $\frac{a+b-(N-2\Delta)}{2}$ and $\frac{a+b-2\Delta}{2}$ will be closer to the minima point $\frac{ab}{N}$. When $\frac{ab}{N} \le \frac{a+b-(N-2\Delta)}{2} \le \frac{a+b-2\Delta}{2}$, $\mathcal{U}_{A,B} = f(\frac{a+b-2\Delta}{2})$. It decreases with the increase of $\Delta$ (or the decrease of $h_A$) because $\frac{a+b-2\Delta}{2}$ gets closer to the minima point $\frac{ab}{N}$. Similarly, when $\frac{a+b-(N-2\Delta)}{2} \le \frac{a+b-2\Delta}{2} \le \frac{ab}{N}$, $\mathcal{U}_{A,B}$ decreases with the increase of $\Delta$ (or the decrease of $h_A$). To sum up, for small $h_A$, the upper bound of $MI(A, B)$, i.e., $\mathcal{U}_{A,B}$, decreases with the decrease of $h_A$.

## C   ALGORITHM

The detailed algorithm for AUTOSSL-ES is shown in Algorithm 1. Concretely, for each round (iteration) of AUTOSSL-ES, we sample $K$ sets of task weights, i.e., $K$ different combinations of SSL tasks, from a multivariate normal distribution. Then we train $K$ graph neural networks independently on each set of task weights. Afterwards, we calculate the pseudo-homohily for each network and adjust the mean and variance of the multivariate normal distribution through CMA-ES based on their pseudo-homohily.

The detailed algorithm for AUTOSSL-DS is summarized in Algorithm 2. Specifically, we first update the GNN parameter $\theta$ through one step gradient descent; then we perform $k$-means clustering to obtain centroids, which are used to calculate the homophily loss $\mathcal{H}$. Afterwards, we calculate the meta-gradient $\nabla_{\{\lambda_i\}}^{\text{meta}}$, update $\{\lambda_i\}$ through gradient descent and clip $\{\lambda_i\}$ to $[0,1]$.

---

**Algorithm 1:** AUTOSSL-ES: AutoSSL with Evolutionary Strategy

---

**for** $r$ *in* $\{0, \ldots, R\}$ **do**
  1. Sample $K$ sets of tasks weights from a multivariate normal distribution
  2. Train $K$ networks w.r.t. each set of task weights from scratch
  3. Calculate pseudo-homophily P-H of node embeddings from each network
  4. Adjust the multivariate normal distribution through CMA-ES based on P-H
**end**

---

**Algorithm 2:** AUTOSSL-DS: AutoSSL with Differential Search

---

Initialize self-supervised task weights $\{\lambda_i\}$ and GNN parameters $\theta$;
**for** $t$ *in* $\{0, \ldots, T\}$ **do**
  1. $\theta_{t+1} = \theta_t - \epsilon \nabla_{\theta_t} \mathcal{L}(f_{\theta_t}, \{\lambda_i, \ell_i\})$
  2. Perform $k$-means clustering on $f_{\theta_t}(\mathcal{G})$ and obtain centroids $\{\mathbf{c}_1, \mathbf{c}_2, \ldots, \mathbf{c}_k\}$
  3. Calculate $p(\mathbf{c}_i \mid \mathbf{x})$ according to Eq. (4)
  4. Calculate homophily loss $\mathcal{H}$ according to Eq. (5)
  5. $\{\lambda_i\} \leftarrow \{\lambda_i\} - \eta \nabla_{\{\lambda_i\}}^{\text{meta}}$
  6. Clip $\{\lambda_i\}$ to [0,1]
**end**

---

## D   DISCUSSIONS ON HOMOPHILY ASSUMPTION

Most of the graphs in our real life satisfy the homophily assumption (McPherson et al., 2001), such as social networks, citation networks, co-purchase networks, etc. Thus, in general, we can treat homophily as a prior knowledge for a majority of real-world graphs. Moreover, it has been shown in (Zhu et al., 2020a; Pei et al., 2020) that most GNNs (such as GCN, GAT, ChebyNet and GraphSage) heavily rely on the homophily assumption and fail to generalize to low-homophily (heterophily) graphs even with label information. Thus, following the design of most GNNs, we focus on the homophily graphs. In addition, to apply our method on heterophily graphs, we can use the graph transformation algorithm (Suresh et al., 2021) to increase the homophily of a given graph. While heterophily graphs also exist in real-world applications, the research of GNNs on heterophily graphs is still at the very early stage even in the cases where the label information is available. Therefore, we will leave the research for heterophily graphs in the unsupervised setting as a future work.

## E   ADDITIONAL EXPERIMENTAL RESULTS

### E.1   RELATIONSHIP BETWEEN DOWNSTREAM PERFORMANCE AND PSEUDO-HOMOPHILY

We provide more results on the relation between downstream performance and pseudo-homophily in Figure 6. Observations are already made in Section 4.4.

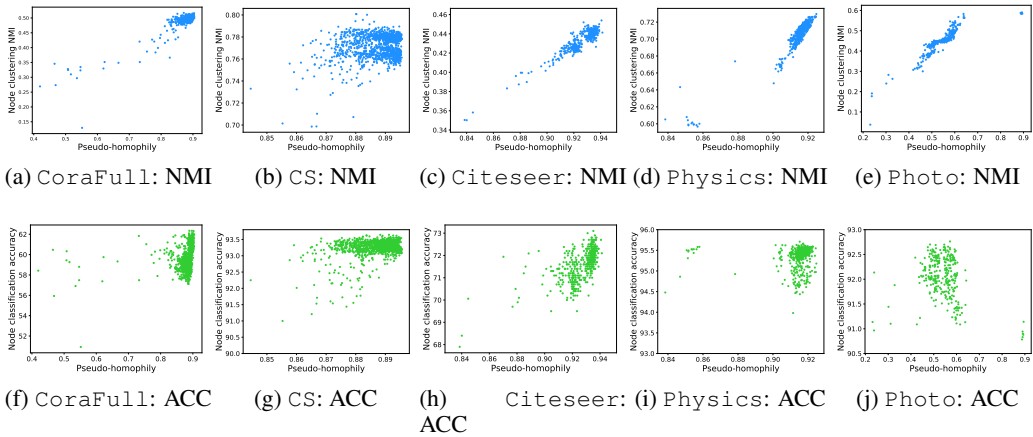

(a) `CoraFull`: NMI    (b) `CS`: NMI    (c) `Citeseer`: NMI (d) `Physics`: NMI    (e) `Photo`: NMI

(f) `CoraFull`: ACC    (g) `CS`: ACC    (h)    `Citeseer`: (i) `Physics`: ACC    (j) `Photo`: ACC
                                                ACC

Figure 6: Relationship between downstream performance and pseudo-homophily.

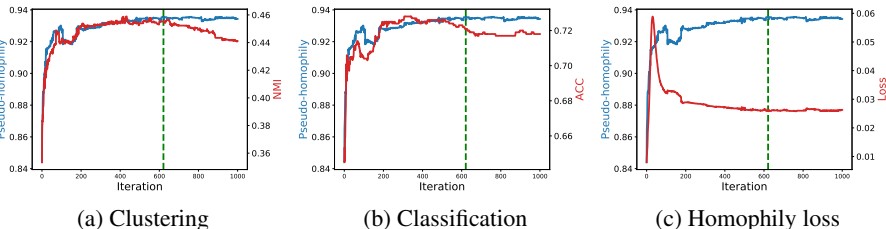

(a) Clustering             (b) Classification             (c) Homophily loss

Figure 7: Pseudo-homophily versus NMI/ACC/Loss on `Citeseer` for AUTOSSL-DS. The vertical dashed line indicates the iteration when pseudo-homophily reaches the maximum value.

### E.2 PSEUDO-HOMOPHILY OVER ITERATIONS

We investigate how pseudo-homophily changes over iterations for AUTOSSL-DS. The changes of pseudo-homophily, NMI, ACC and homophily loss (Eq. (5)) are plotted in Figure 7. From Figure 7a and 7b, we can observe that pseudo-homophily first increases and then becomes stable through iterations. The situation is a bit different for clustering and classification performance: NMI and ACC first increase with the increase of pseudo-homophily and then drop when pseudo-homophily is relatively stable. This indicates that overtraining can hurt downstream performance as the model will have the risk of overfitting on the combined SSL tasks. However, as shown in the figure, if we stop at the iteration when pseudo-homophily reaches the maximum value we can still get a high NMI and ACC. On a separate note, Figure 7c shows how the homophily loss used in AUTOSSL-DS changes over iterations. We note that in the first iterations the homophily loss is low but the pseudo-homophily is also low. This is because the embeddings in the first few epochs are less separable and would lead to very close soft-assignment of clusters. As shown in the figure, however, the problem is resolved as the embeddings become more distinguishable through iterations. Thus, we argue that the homophily loss in Eq. (5) is still a good proxy in optimizing pseudo-homophily.

### E.3 EFFICIENCY ANALYSIS

#### E.3.1 TIME COMPLEXITY ANALYSIS

We analyze the time complexity of the proposed AUTOSSL. Here we call one set of task weights $\{\lambda_i\}$ as one candidate solution. We denote the time of training one epoch on a given set of SSL tasks as $t_o$ and the evaluation time is $t_e$. Suppose we need to train $T$ epochs for the network. Then the time for running one single candidate solution is $Tt_o + t_e$; the time of running $R$ rounds of AUTOSSL-ES should be $RTt_o + Rt_e$. For AUTOSSL-DS, the running time is $Tt_o + Tt_e$. As an illustration, in an $L$-layer GCN with $d$ as the number of hidden dimensions, $t_o$ can be expressed as $O(L|\mathcal{E}|d + LNd^2)$ and $t_e$ has time complexity of $O(KINd)$ with $K$ being the number of clusters and $I$ being the

number of iterations for $k$-means. Hence, we also express the time complexity of AUTOSSL-ES as $O(RTL|\mathcal{E}|d + RTLNd^2 + RKINd)$ and that of AUTOSSL-DS as $O(TL|\mathcal{E}|d + TLNd^2 + TKINd)$. Both of them linearly increase with the number of nodes $N$ when $\mathcal{E}$ is proportional to $N$. We note that in AUTOSSL-DS, the complexity of calculating the second-order derivatives in backward propagation has an additional factor of $O(|\theta||\{\lambda_i\}|)$, which can be reduced to $O(|\{\lambda_i\}|)$ with approximated Hessian-vector products. The factor can be neglected as the number of tasks $O(|\{\lambda_i\}|)$ is small.

### E.3.2 Empirical Comparison

For empirical comparison, we take `Citeseer`, `Photo`, `CoraFull` as examples to illustrate. As shown in Table 5, we compare the running time of different methods for training 1000 epochs on one NVIDIA-K80 GPU. The column of 5-Tasks indicates the running time of training a combination of 5 SSL tasks, i.e. CLU, PAR, PAIRSIM, PAIRDIS and DGI, for 1000 epochs. Note that we report the running time of AUTOSSL-ES as the multiplication between 500 and the time of 5-Tasks, i.e., running 500 candidate solutions. From the table, we can see that the running time of AUTOSSL-ES depends on the number of candidate solutions and usually takes a long time to run. However, AUTOSSL-DS significantly reduces the running time of AUTOSSL-ES. It is worth noting the state-of-the-art SSL task, MVGRL, takes a long time to run and suffers from the OOM issue when the dataset gets larger.

Table 5: Comparison of running time for training 1000 epochs on one NVIDIA-K80 GPU (12 GB memory). OOM indicates out-of-memory on this GPU.

|  | DGI | MVGRL | 5-Tasks | AUTOSSL-ES | AUTOSSL-DS |
|---|---|---|---|---|---|
| Citeseer | 222s | 1220s | 322s | 322s×500 | 1222s |
| Photo | 177s | 1074s | 507s | 507s×500 | 1766s |
| CoraFull | 553s | OOM | 858s | 858s×500 | 3584s |

## F Comparison with Different Strategy

In this subsection, we examine how other strategies of assigning task weights affect the quality of learned representations. The results are summarized in Table 6. In this table, "Best SSL" indicates the best performance achieved by the individual tasks; "Random Weight" indicates the performance achieved by randomly assigning task weights; "Equal Weight" indicates the performance achieved by assigning the same task weights (i.e., all 1). The values that outperform "Best SSL" are under-lined. From the table, we make two observations. **Obs 1.** Unlike AUTOSSL, "Random Weight" and "Equal Weight" would hurt both NMI and ACC on some datasets, e.g., `Citeseer`. This suggests that SSL tasks might conflict with each other and thus harm the downstream performance. **Obs 2.** In some cases like `Physics`, "Equal Weight" can also improve both ACC and NMI, which aligns well with our initial motivation that combinations of SSL can help capture multiple sources of information and benefit the downstream performance. The two observations suggest that it is important to design a clever strategy that can automatically compose graph SSL tasks.

## G Comparison with Random Search

In this subsection, we choose `Citeseer` to study the difference between random search and evolutionary algorithm (AUTOSSL-ES), and report the result in Table 7. Specifically, the random search method randomly generates 800 sets of tasks weights and we evaluate the pseudo-homophily from the models trained with those task weights. Note that in AUTOSSL-ES we also evaluated 800 sets of tasks weights in total. From the table, we can see that random search is not as efficient as AUTOSSL-ES: with the same search cost, the resulted pseudo-homophily of random search is not as high as AUTOSSL-ES and the downstream performance is also inferior. This result suggests that search with evolutionary algorithm can find the optimum faster than random search.

Table 6: Performance comparison of different strategies of assigning task weights. The NMI rows indicate node clustering performance; ACC rows indicate node classification accuracy (%); P-H stands for pseudo-homophily. (Underline: better than "Best SSL").

| Dataset | Metric | Best SSL | Random Weight | Equal Weight | AUTOSSL-ES | AUTOSSL-DS |
|---------|--------|----------|---------------|--------------|------------|------------|
| Citeseer | NMI | $0.439_{\pm 0.00}$ | $0.398_{\pm 0.01}$ | $0.408_{\pm 0.00}$ | $0.449_{\pm 0.01}$ | $0.449_{\pm 0.01}$ |
|          | ACC | $71.64_{\pm 0.44}$ | $70.64_{\pm 0.07}$ | $70.80_{\pm 0.31}$ | $72.14_{\pm 0.41}$ | $72.00_{\pm 0.32}$ |
|          | P-H | — | $0.897$ | $0.904$ | $0.943$ | $0.934$ |
| Computers | NMI | $0.433_{\pm 0.00}$ | $0.341_{\pm 0.03}$ | $0.290_{\pm 0.00}$ | $0.447_{\pm 0.01}$ | $0.448_{\pm 0.01}$ |
|           | ACC | $87.26_{\pm 0.15}$ | $86.86_{\pm 0.25}$ | $87.24_{\pm 0.38}$ | $87.26_{\pm 0.64}$ | $88.18_{\pm 0.43}$ |
|           | P-H | — | $0.406$ | $0.378$ | $0.503$ | $0.511$ |
| CoraFull | NMI | $0.498_{\pm 0.00}$ | $0.458_{\pm 0.02}$ | $0.493_{\pm 0.00}$ | $0.506_{\pm 0.01}$ | $0.500_{\pm 0.00}$ |
|          | ACC | $60.42_{\pm 0.39}$ | $58.88_{\pm 0.32}$ | $59.01_{\pm 0.29}$ | $61.01_{\pm 0.50}$ | $61.10_{\pm 0.68}$ |
|          | P-H | — | $0.811$ | $0.868$ | $0.903$ | $0.895$ |
| CS | NMI | $0.767_{\pm 0.01}$ | $0.761_{\pm 0.01}$ | $0.770_{\pm 0.01}$ | $0.772_{\pm 0.01}$ | $0.771_{\pm 0.01}$ |
|    | ACC | $92.75_{\pm 0.12}$ | $92.88_{\pm 0.20}$ | $93.22_{\pm 0.12}$ | $93.26_{\pm 0.16}$ | $93.35_{\pm 0.09}$ |
|    | P-H | — | $0.879$ | $0.881$ | $0.895$ | $0.890$ |
| Photo | NMI | $0.509_{\pm 0.01}$ | $0.341_{\pm 0.02}$ | $0.366_{\pm 0.02}$ | $0.560_{\pm 0.04}$ | $0.511_{\pm 0.03}$ |
|       | ACC | $92.08_{\pm 0.37}$ | $92.04_{\pm 0.28}$ | $92.54_{\pm 0.29}$ | $92.04_{\pm 0.89}$ | $92.71_{\pm 0.32}$ |
|       | P-H | — | $0.412$ | $0.472$ | $0.791$ | $0.626$ |
| Physics | NMI | $0.704_{\pm 0.00}$ | $0.692_{\pm 0.00}$ | $0.709_{\pm 0.01}$ | $0.725_{\pm 0.00}$ | $0.726_{\pm 0.00}$ |
|         | ACC | $95.07_{\pm 0.06}$ | $95.09_{\pm 0.08}$ | $95.39_{\pm 0.10}$ | $95.57_{\pm 0.02}$ | $95.13_{\pm 0.36}$ |
|         | P-H | — | $0.914$ | $0.916$ | $0.921$ | $0.923$ |
| WikiCS | NMI | $0.341_{\pm 0.01}$ | $0.305_{\pm 0.01}$ | $0.323_{\pm 0.01}$ | $0.366_{\pm 0.01}$ | $0.344_{\pm 0.02}$ |
|        | ACC | $75.81_{\pm 0.17}$ | $76.29_{\pm 0.17}$ | $76.49_{\pm 0.21}$ | $76.80_{\pm 0.13}$ | $76.58_{\pm 0.28}$ |
|        | P-H | — | $0.675$ | $0.690$ | $0.751$ | $0.749$ |

Table 7: Comparison with random search. The NMI indicates node clustering performance; ACC indicates node classification accuracy (%); P-H stands for pseudo-homophily.

| Dataset | Metric | Random Search | AUTOSSL-ES |
|---------|--------|---------------|------------|
| Citeseer | NMI | $0.443_{\pm 0.00}$ | $0.449_{\pm 0.01}$ |
|          | ACC | $71.68_{\pm 0.55}$ | $72.14_{\pm 0.41}$ |
|          | P-H | $0.934$ | $0.943$ |

## H  BROADER IMPACT

Graph neural networks (GNNs) are commonly used for node and graph representation learning tasks due to their representational power. Such models have also been recently proposed for use in large-scale social platforms for tasks including forecasting (Tang et al., 2020a), friend ranking (Sankar et al., 2021) and item recommendation (Ying et al., 2018; Wu et al., 2019a), which impact many end users. Like other machine learning models, GNNs can suffer from typical unfairness issues which may arise due to sensitive attributes, label parity issues, and more (Dai & Wang, 2021). Moreover, GNNs can also suffer from degree-related biases (Tang et al., 2020b). Self-supervised learning (SSL) is often used to learn high-quality representations without supervision from labeled data sources, and is especially useful in low-resource settings or in pre-training/fine-tuning scenarios. Several works have illustrated the potential for representations learned in a self-supervised way to encode bias unintentionally, for example in language modeling (Bender et al., 2021; Zhao et al., 2019), image representation learning (Roberts et al., 2018) and outlier detection (Shekhar et al., 2020).

Our work on automated self-supervised learning with graph neural networks shares the caveats of these two domains in terms of producing inadvertent or biased outcomes. We propose an approach to learn self-supervised representations by utilizing multiple types of pretext tasks in conjunction with one another. While this produces improved performance on standard tasks used for benchmarking representation quality, it does not guarantee that these representations are fair and should be used without typical fairness checks in industrial contexts. However, such concerns are not inherently posed by our proposed ideas, but by the foundations it builds on in GNNs and SSL. We anticipate our ideas will drive further research in more sophisticated and powerful self-supervised graph learning, and do not anticipate direct negative outcomes from this work.

