# OpenReview forum: "Automated Self-Supervised Learning for Graphs"
_ICLR.cc/2022/Conference — ICLR 2022 Poster_

### Official Review · Reviewer_DNwN · 2021-11-02

**Correctness:** 4
**Technical Novelty And Significance:** 2
**Empirical Novelty And Significance:** 2
**Recommendation:** 6
**Confidence:** 3

**Main Review:**


1. "Is Homophily a Necessity for Graph Neural Networks?" is a paper. In that paper, it states Homophily is not necessity. If dataset is Homophily, the task comes to easy. In this paper, authors use such easy datasets.
2. If the meta-learning is the contribution, it is not a contribution of the paper but is a tool.
3. The experimental results show it obtains better performance.
4. Is the "evolutionary strategy" a contribution?


**Summary Of The Paper:**

Authors use homophily datasets and meta learning. For self-supervised learning, authors use evolutionary strategy.

**Summary Of The Review:**

 homophily datasets is used and authors shows theoretical analysis.

---

> ### Author Response · Authors · 2021-11-19
> **Response to Reviewer Reviewer DNwN**
>
> We appreciate your efforts in reviewing our paper and your constructive questions about our work. Please see our detailed responses to your comments below.
>
> ---
> Q1. "Is Homophily a Necessity for Graph Neural Networks?" is a paper.  In that paper, it states Homophily is not a necessity. If the dataset is Homophily, the task comes to easy.  In this paper, authors use such easy datasets.
>
> A1.  We would like to draw your attention to the following points:
> * Firstly, most of the graphs in our real life satisfy the homophily assumption [1], such as social networks, citation networks, co-purchase networks, etc. Thus, in general, we can often treat homophily as a reasonable prior  for a majority of real-world graphs.
> * Second, it has been shown in [2] that most GNNs (such as GCN, GAT, ChebyNet and GraphSage) heavily rely on the homophily assumption and fail to generalize to low-homophily (heterophily) graphs even with label information -- the paper you mentioned also shows this to be true in general (although certain heterophily cases can be handled well by such models). Thus, following the design of most GNNs, we focus on the homophily graphs.
> * Third, to the best of our knowledge, almost all the previous works on node-level SSL focus only on homophily graphs [3,4,5,6].  Since our goal is to combine these SSL tasks, it is natural to use those homophily datasets.
> * Fourth, to apply our method on heterophily graphs, we can envision using the graph transformation algorithm, designed in the recent work [7], to increase the homophily of a given graph.
>
> We had included the above discussion in Appendix  D.
>
> ----
>
> Q2. If the meta-learning is the contribution, it is not a contribution of the paper but is a tool.
>
> A2. The meta-learning itself is not a contribution of this paper. Our contribution is the proposed AutoSSL framework under the unsupervised paradigm which optimizes a novel concept of pseudo-homophily through meta-gradient descent (DS) or evolution strategy (ES).  The detailed contributions of AutoSSL-DS can be summarized as:
> * **Tackling the challenge of the unsupervised setting.** The unsupervised setting is highly challenging in that we do not have an easy-to-get objective to guide the search process, e.g., the validation loss as in traditional NAS papers. To address this issue, we propose a novel concept of pseudo-homophily in the absence of ground truth labels and show that optimizing pseudo-homophily can help learn generalizable node representations from both theoretical and empirical perspectives. This property is exclusive for graphs and most AutoML works do not aim to solve the unsupervised AutoML problem.
> * **Optimization strategy.** First, the pseudo-homophily involves a hard assignment process during clustering, which poses challenges in updating task weights with gradient descent. To address this issue, we convert the hard clustering into soft clustering and optimize the homophily loss in Eq (5).  Second, we cannot use simple gradient descent to update the task weights as homophily loss does not directly involve the task weights. Hence, we employ meta-gradient descent to optimize the homophily loss, which completes our framework of AutoSSL-DS.
>
>  ----
> Q3. Is the "evolutionary strategy" a contribution?
>
> A3. The evolutionary strategy itself is not a contribution of this paper.  Our contribution involves the formulation of automated graph SSL and the proposal together with the justification of pseudo-homophily, which is shown to maximize the upper bound of mutual information between pseudo-labels and downstream labels. To optimize the pseudo-homophily, AutoSSL-ES is a solution we provided to guide the unsupervised search process and is demonstrated to learn generalizable node representations for various datasets and different downstream tasks.
>
>
> In light of these responses, we hope we have addressed your concerns.  If we have left any notable points of concern unaddressed, please do share and we will attend to these points.
>
> ----
>
> [1] Birds of a feather: Homophily in social networks. Annual review of sociology, 2001.\
> [2] Beyond homophily in graph neural networks: Current limitations and effective designs. NeurIPS, 2020.\
> [3] Self-supervised learning of graph neural networks: A unified review. arXiv, 2021.\
> [4] When does self-supervision help graph convolutional networks? ICML, 2020.\
> [5] Contrastive multi-view representation learning on graphs. ICML, 2020.\
> [6] Graph contrastive learning with adaptive augmentation. WWW, 2021.\
> [7]  Breaking the Limit of Graph Neural Networks by Improving the Assortativity of Graphs with Local Mixing Patterns. KDD, 2021.

---

### Official Review · Reviewer_yXJg · 2021-11-02

**Correctness:** 3
**Technical Novelty And Significance:** 2
**Empirical Novelty And Significance:** 2
**Recommendation:** 5
**Confidence:** 4

**Details Of Ethics Concerns:**

None.

**Main Review:**


(1) Performance gains are small or non-significant. Closely examining Tables 1-3, I see many scenarios where the best performing AutoSSL model performs on par with the strongest baselines or offers minuscule improvements. (that is, when taking into account standard variation of performance across independent runs). For example, in Table 2, results for CoraFull are 61.10±0.68 vs. 60.56±0.33 (= on-par performance), results for Physics are 95.57±0.02 vs. 94.66±0.10 (= minuscule improvement, which we only see after considering the stronger AutoSSL-ES variant, AutoSSL-DS performs at 95.13±0.36, which leads to essentially no performance). There are many such scenarios in Tables 1-3.

(2) The AutoSSL approach is concerned with finding an effective way to combine individual SSL tasks (problem definition in Eq. 1, where $\mathcal{H}$ is set to the negative pseudo-homophily). That is, it assumes that a set of $n$ SSL tasks are already given and it will find a good combination of those tasks formulated as a weighted addition of individual SSL tasks, each task associated with a learned task weight $\lambda_i$. Such formulation seems quite restrictive. It does not allow for any interaction between the loss functions. Further, it also assumes that the user already knows what are potentially good SSL tasks.

(3) It is unclear how the proposed strategy could be used for other contexts, such as link prediction and graph classification.

**Summary Of The Paper:**

Self-supervised learning (SSL) methods for graphs take a given graph with node attribute information and construct various SSL tasks using structural and attribution information. These tasks provide self-supervision for training graph neural networks without accessing any labeled data. Existing studies show that different SSL tasks can lead to varying downstream performance across tasks, suggesting that the success of SSL tasks strongly depends on the dataset and the downstream task. This observation is important, yet not surprising, as similar findings were made in other data modalities (text, vision). As a result, selecting the 'right' SSL tasks can be important, which serves as a motivation for this study.

This study develops an approach (called AutoSSL) for combining multiple SSL tasks for unlabeled representation learning. To this end, it defines a pseudo-homophily metric to measure the quality of learned representations. Based on the pseudo-homophily, the study describes two techniques to search & combine SSL tasks, one using an evolutionary algorithm and the other using meta-gradient descent. The AutoSSL approach is evaluated on eight datasets considering combinations of five SSL tasks (one contrastive learning task and four predictive tasks).

**Summary Of The Review:**

Both AutoSSL variants (evolutionary strategy and the gradient descent version) seem ad-hoc and straightforward combinations of existing tools. This is not a problem on its own, however, the empirical gains are not convincing.

---

> ### Author Response · Authors · 2021-11-19
> **Response to Reviewer yXJg - Part 1**
>
> Thank you for your detailed comments and constructive suggestions. To address your concerns, we provide the following responses.
>
> ---
> Q0. AutoSSL variants seem ad-hoc and straightforward combinations of existing tools.
>
> A0. (1) Firstly, we would like to highlight that **the proposed AutoSSL variants are not ad-hoc**.  AutoSSL aims to maximize the proposed pseudo-homophily, which is shown to maximize the upper bound of mutual information between pseudo-labels and downstream labels ( see Theorem 1). From both theoretical and empirical perspectives, it is shown to be a good surrogate measure for mutual information and yields good performances on various real-world datasets.
>
> (2) AutoSSL-ES presents an intuitive solution towards optimization of pseudo-homophily but it is less efficient; AutoSSL-DS is more efficient as it is gradient based search. However, the pseudo-homophily involves a hard assignment process during clustering, which poses challenges in updating task weights with gradient descent. To address this issue, we spend efforts in converting the hard clustering into soft clustering which leads to the final homophily loss in Eq (5).
>
> Together with the pseudo-homophily tailed for the unsupervised setting, we argue that these efforts are not "straightforward combinations of existing tools".
>
> ---
> Q1. Performance gains are small or non-significant.  For example, in Table 2, results for CoraFull are 61.10±0.68 vs. 60.56±0.33 (= on-par performance), results for Physics are 95.57±0.02 vs. 94.66±0.10 (= minuscule improvement, which we only see after considering the stronger AutoSSL-ES variant, AutoSSL-DS performs at 95.13±0.36, which leads to essentially no performance).
>
> A1.  We kindly point out that the combined SSL is able to achieve good performance *on various downstream tasks as well as different datasets*, while existing methods fail to achieve this goal:
>
> * **Maintaining good performance on different downstream tasks.** Greatly improving one of the downstream tasks (clustering or classification) while maintaining the performance on the other task can be considered a significant improvement. Let’s take a look at the example of CoraFull you mentioned in the main review. In Table 2, on node classification AutoSSL vs. MVGRL is 61.10±0.68 vs. 60.56±0.33 (=mild improvement), while on node clustering AutoSSL vs. MVGRL is 0.506±0.01 vs. 0.400±0.01 (=more than 25% relative improvement). The performance of MVGRL detracts from our goal of *learning generalizable node representations* that can benefit different downstream tasks. Similar significant improvements from AutoSSL can be found in other cases in Table 1, 2 and 3.
> * **Average rank over different datasets.** AutoSSL is also able to achieve the best performance across different datasets. To see it clearly, we now compute the average ranks of all the methods over eight datasets on clustering and classification and report them in the following two tables (we have also included them in the main content of our paper). We can see that the two AutoSSL variants consistently achieve the best ranks on both clustering and classification.
>
> |                |     |     |         |         |     | AutoSSL | AutoSSL    |
> |----------------|-----|-----|---------|---------|-----|:-------:|:---:|
> |                | Clu | Par | PairSim | PairDis | DGI | ES      | DS  |
> | Clustering     | 6.3 | 3.5 | 4.1     | 6.5     | 4.6 | 1.3     | 1.5 |
> | Classification | 6.8 | 3.9 | 4.9     | 5.4     | 3.9 | 1.8     | 1.4 |
>
> |     |       |         |     |      |       |       | AutoSSL | AutoSSL |
> |----|---|----|---|---|---|:-----:|:-----:|-----|
> |      | Init. | RawFeat | GAE | VGAE | ARVGA | MVGRL | ES      | DS      |
> | Clustering     | 6.4   | 7.1     | 4.3 | 4.0  | 6.4   | 4.3   | 1.6     | 2.1     |
> | Classification | 6.0   | 7.0     | 4.1 | 4.6  | 7.1   | 3.1   | 1.9     | 1.5     |
>
> We also included the significance results in Tables 1-3 from paired t-tests to justify the improvement from our framework:
>
> * **Significance test.** We used the blue and red numbers to indicate the statistically significant improvements over the best individual task, via paired t-tests with significance levels  of 0.05 and 0.1, respectively. Specifically, p-value indicates the probability that the significance of the results occurred by chance.  As we can see from the tables, the improvements from AutoSSL *over the best baselines on each task* are statistically significant at level 0.05 in most cases.

---

> > ### Author Response · Authors · 2021-11-19
> > **Response to Reviewer yXJg - Part 2**
> >
> > ---
> >
> > Q2. The AutoSSL approach will find a good combination of those tasks formulated as a weighted addition of individual SSL tasks, each task associated with a learned task weight. Such formulation seems quite restrictive. It does not allow for any interaction between the loss functions.
> >
> > A2.  The formulation (combining multiple loss functions) is not restrictive because *there do exist interactions between the loss functions*. Specifically, the GNN model is trained on the combination of the SSL losses. Optimizing the combined loss will impact the model parameters and thus affect the loss values of all SSL tasks.  Such formulation has also been demonstrated effective in improving supervised tasks such as face recognition [1,2],  machine translation [3] and recommender systems [4].
> >
> >
> > ---
> >
> > Q3. Further, it also assumes that the user already knows what are potentially good SSL tasks.
> >
> > A3. We agree that we should not assume what are potentially good SSL tasks since we do not have such knowledge in the unsupervised setting. **That is exactly why the studied problem is challenging corresponding to our major contribution because we do not have the assumption that the user knows what are potentially good SSL tasks**. Instead, AutoSSL is able to remove the effect of some bad SSL tasks by lowering the weights of those bad tasks. As we can see  from Table 1, Clu generally performs very badly on both classification and clustering. From the visualized weights in Figure 3, we observe that the weight of Clu searched by AutoSSL is extremely small (close to 0), indicating that Clu is not very helpful on those datasets and AutoSSL can avoid its negative effect.
> >
> > ---
> >
> > Q4.  It is unclear how the proposed strategy could be used for other contexts, such as link prediction and graph classification.
> >
> > A4. (1) Since link prediction also requires learning node-level representation, it is straightforward to apply AutoSSL in link prediction. Concretely, we can first use AutoSSL to learn the node embedding and then we can either use the inner product of node embeddings to predict edges or learn a parameterized score function to predict edges given node embeddings.
> >
> > (2) We can also easily extend the AutoSSL framework to graph classification: we first employ AutoSSL to learn node embeddings for each graph and apply flat pooling layers on node embeddings to obtain graph representation.
> >
> > We leave the empirical exploration of AutoSSL on link prediction and graph classification as future work.
> >
> > ---
> >
> > In light of these responses, we hope we have addressed your concerns, and hope you will consider increasing your score.  If we have left any notable points of concern unaddressed, please do share and we will attend to these points.
> >
> > ---
> >
> > [1]  Chuming Li, Xin Yuan, Chen Lin, Minghao Guo, Wei Wu, Junjie Yan, and Wanli Ouyang.  Am-lfs:Automl for loss function search.  ECCV 2019.\
> > [2] Xiaobo Wang, Shuo Wang, Cheng Chi, Shifeng Zhang, and Tao Mei.  Loss function search for face recognition. ICML 2020.\
> > [3] Haowen Xu, Hao Zhang, Zhiting Hu, Xiaodan Liang, Ruslan Salakhutdinov, and Eric Xing.  Autoloss: Learning discrete schedules for alternate optimization. ICLR 2019.\
> > [4] Xiangyu Zhao, Haochen Liu, Wenqi Fan, Hui Liu, Jiliang Tang, and Chong Wang.  Autoloss:  Automated loss function search in recommendations.  KDD 2021.

---

### Official Review · Reviewer_H8HY · 2021-11-03

**Correctness:** 4
**Technical Novelty And Significance:** 2
**Empirical Novelty And Significance:** 2
**Recommendation:** 8
**Confidence:** 4

**Main Review:**

Strengths:
The paper is well-written and easy to follow. I can easily understand the idea of this paper.

This paper conducts extensive experiments on 7 different datasets from different domains to show their improved performances, which is very reasonable and convincing to me.

Weaknesses:
The main concern I have towards this paper is its novelty. I can see there is much effort of this paper to make adjusting different pretext tasks on a graph, especially the authors bring up the important "homophily" property of graph to create pseudo labels for self-supervision. However, the essence of this paper is still dynamically adjusting weights for different losses, as the authors mentioned in the section Related Work of Automated Loss Function Search. There is so much work on reweighting different loss functions which have been well studied. Though the authors mentioned that "the problem of self-supervised loss search for graphs remains rarely explored", I find out the solved problem in this paper is actually an old problem, which is limited in novelty and significance.

The core contributions of this paper mainly include (i) making use of homophily to create pseudo labels for self-supervision; (ii) proposing AutoSSL-ES based evolutionary strategy (iii) proposing AutoSSL-DS based on meta-gradient descent. For (i), after checking the paper and appendix, I think it is good and reasonable. But why should the authors propose two different methods to solve the problem? The author mentions AutoSSL-ES requires evaluating a large population of candidate combinations which is not practical and AutoSSL-DS is much efficient. So, what are the advantages of AutoSSL-ES? The authors should give more explanations and analysis towards choosing these two methods. Otherwise, I would consider they are simply a combination of existing works without much contribution.

I observe that the improvement of AutoSSL compared with the best result of individual tasks is not that significant. Moreover, the best results of individual tasks are from PAR and DGI. This leads to a question that does is really needs to weigh these 5 tasks. Maybe considering the other 3 tasks is not that helpful.

Also, more recent works of graph self-supervised learning can be considered and cited. For example, Han X, Huang Z, An B, et al. Adaptive Transfer Learning on Graph Neural Networks[C]//Proceedings of the 27th ACM SIGKDD Conference on Knowledge Discovery & Data Mining. 2021: 565-574.

**Summary Of The Paper:**

This paper mainly studies the problem of automatically weighting multiple self-supervised tasks on a graph without information of true labels, in order to make downstream node-level prediction tasks perform better. To overcome the challenge of missing ground truth in pre-training tasks, the authors propose to use a property named "homophily" of the graph to create pseudo labels. Then, they propose two algorithms to automatically adjust weights among different tasks: one is based on evolution algorithms; another is based on gradient descent. Extensive experiments on different real-world datasets show the improvement of the proposed methods compared with a single pretext task.

**Summary Of The Review:**

Though this paper is well-written and studies an important and popular problem in graph learning. I still have concerns in different aspects.

---

> ### Author Response · Authors · 2021-11-19
> **Response to Reviewer H8HY  (1/3)**
>
> We appreciate the reviewer's perception of our contributions and thank the reviewer for the insightful questions. Our detailed responses are below.
>
> ----
>
> Q1. The novelty of this work.
>
> A1. To enable automated self-supervised learning on graphs, we face significant challenges that call for novel solutions. Below we highlight several key challenges and innovations in our work:
>
> (1) **The challenge of the unsupervised setting.** The unsupervised setting is highly challenging in that we do not have an easy-to-get objective to guide the search process, e.g., the validation loss as in traditional NAS or loss function search. To address this issue, we take advantage of an essential property of many real-world graphs, i.e. homophily, to guide the search process. This property is *exclusive for graphs* and most works on AutoML do not aim to solve the *unsupervised* AutoML problem.
>
> (2) **The justification of the proposed pseudo-homophily.** To address the aforementioned challenge, we proposed to use pseudo-homophily as the search guidance. We show that increasing pseudo-homophily can indeed help learn better node representations from both theoretical and empirical perspectives.
>
> (3) **Optimization strategy.** First, the pseudo-homophily involves a hard assignment process during clustering, which poses challenges in updating task weights with gradient descent. To address this issue, we convert the hard clustering into soft clustering and optimize the homophily loss proposed in Eq (5).  Second, we cannot use simple gradient descent to update the task weights as homophily loss does not directly involve the task weights. This also makes our work different from traditional NAS works as their outer objective (e.g., validation loss) directly involves the model architecture which is the search goal.
>
> (4) **Necessity of Automated SSL.** Without the knowledge of downstream tasks, we do not know in reality which SSL tasks are most suited for transfer.  Our empirical results demonstrate that different SSL tasks can have distinct impact on different datasets and different tasks. Thus, it is important to take advantage of different SSL tasks while automatically controlling their contributions in a principled way.
>
> ---
>
> Q2. What are the advantages of AutoSSL-ES?
>
> A2. AutoSSL-DS and -ES are two solutions we provided to optimize the proposed pseudo-homophily. DS is more efficient but it requires calculating the meta-gradients. Since evolutionary algorithms are black-box optimization tools, ES provides an alternative solution to search SSL task weights when it is hard to compute the meta-gradients, e.g., when the optimization goal is non-differentiable. In addition, ES is a mature global optimization tool given the large usage of time [1] and we found that ES tends to produce higher or comparable pseudo-homophily than DS. It is worth noting that our major contributions focus on the formulation of automated graph SSL and the proposal together with the justification of pseudo-homophily which can then be optimized by ES/DS to guide the unsupervised search process.
>
> [1] AutoML: A Survey of the State-of-the-Art. Knowledge-Based Systems, Vol. 215, Jan 2021.

---

> > ### Author Response · Authors · 2021-11-19
> > **Response to Reviewer H8HY  (2/3)**
> >
> > ---
> >
> > Q3. I observe that the improvement of AutoSSL compared with the best result of individual tasks is not that significant.
> >
> > A3.  Thanks for your suggestion -- we have included the significance results in Tables 1-3 from paired t-tests to justify the improvement from our framework:
> > * **Significance test.** We used the blue and red numbers to indicate the statistically significant improvements over the best individual task, via paired t-tests with significance levels  of 0.05 and 0.1, respectively. Specifically, p-value indicates the probability that the significance of the results occurred by chance.  As we can see from the tables, the improvements from AutoSSL *over the best baselines on each task* are statistically significant at level 0.05 in most cases.
> >
> > Furthermore, as our goal is to learn generalizable node representations, we also clarify that the combined SSL can achieve good performance *on various downstream tasks as well as different datasets*, while existing methods fail to achieve this goal:
> >
> > * **Maintaining good performance on different downstream tasks.** Greatly improving one of the downstream tasks (clustering or classification) while maintaining the performance on the other task can be considered a significant improvement. Let’s take a look at the example of CoraFull. In Table 2, on node classification AutoSSL vs. MVGRL is 61.10±0.68 vs. 60.56±0.33 (=mild improvement), while on node clustering AutoSSL vs. MVGRL is 0.506±0.01 vs. 0.400±0.01 (=more than 25% relative improvement). Similar significant improvements from AutoSSL can be found in other cases in Table 1, 2 and 3.
> > * **Average rank over different datasets.** AutoSSL is also able to achieve the best performance across different datasets. To see it clearly, we now compute the average ranks of all the methods over eight datasets on clustering and classification and report them in the following two tables (we have also included them in the main content of our paper). We can see that the two AutoSSL variants consistently achieve the best ranks on both clustering and classification.
> >
> > |                |     |     |         |         |     | AutoSSL | AutoSSL    |
> > |--|--|--|--|--|--|:--:|:--:|
> > |                | Clu | Par | PairSim | PairDis | DGI | ES      | DS  |
> > | Clustering     | 6.3 | 3.5 | 4.1     | 6.5     | 4.6 | 1.3     | 1.5 |
> > | Classification | 6.8 | 3.9 | 4.9     | 5.4     | 3.9 | 1.8     | 1.4 |
> >
> > |                |       |         |     |      |       |       | AutoSSL | AutoSSL |
> > |--|--|--|--|--|--|:--:|:--:|--|
> > |                | Init. | RawFeat | GAE | VGAE | ARVGA | MVGRL | ES      | DS      |
> > | Clustering     | 6.4   | 7.1     | 4.3 | 4.0  | 6.4   | 4.3   | 1.6     | 2.1     |
> > | Classification | 6.0   | 7.0     | 4.1 | 4.6  | 7.1   | 3.1   | 1.9     | 1.5     |
> >
> > Q4. Moreover, the best results of individual tasks are from PAR and DGI. This leads to a question that really needs to weigh these 5 tasks. Maybe considering the other 3 tasks is not that helpful.
> >
> > A4. First, we kindly point out that it is important to design an automated framework that can adaptively control the contributions of different SSL tasks.
> >
> > * **Different datasets and tasks “prefer” different SSL tasks.** For example, in the ranking table shown in A4, DGI performs better than Par in classification while it underperforms ParSim in clustering. This inspires us to develop a framework that can mitigate the effects of bad SSLs and promote good SSLs.
> >
> > * Under the setting of unsupervised (self-supervised) learning, **we cannot access the label information** that can be used to evaluate SSL tasks directly. Thus, we do not have knowledge about the downstream performance of different SSL tasks in advance. It is desired to automatically adjust the contributions of different SSLs.
> >
> > Second, considering the tasks other than Par and DGI can still be helpful. In Figure 3(a), we had shown the searched task weights and we also show them in the following table.
> >
> > |            | Clu  | Par  | PairSim | PairDis | DGI  |
> > |--|--|--|--|--|--|
> > | Physics    | 0.01 | 0.20 | 0.04    | 0.01    | 0.98 |
> > | Photo      | 0.00 | 0.88 | 0.39    | 0.01    | 0.01 |
> > | CS         | 0.00 | 0.42 | 0.41    | 0.00    | 0.95 |
> > | Wiki-CS    | 0.00 | 0.96 | 0.92    | 0.55    | 0.07 |
> > | Citeseer   | 0.02 | 0.04 | 0.20    | 0.18    | 0.99 |
> > | CoraFull   | 0.01 | 0.77 | 0.99    | 0.10    | 0.98 |
> > | Computers  | 0.00 | 1.00 | 0.05    | 0.09    | 0.57 |
> >
> > We can see that the searched task weights of PairSim and PairDis are not always extremely low. For example, the weights for PairSim and PairDis are [0.92, 0.55] on Wiki-CS. This suggests that combining more than the tasks of Par and DGI can be even more helpful. We also observe that the weights of Clu are always very small across datasets. It could be the reason that Clu is generally harmful to the downstream tasks of those datasets and AutoSSL is able to remove its effect by decreasing its task weight.

---

> > > ### Author Response · Authors · 2021-11-19
> > > **Response to Reviewer H8HY  (3/3)**
> > >
> > > ---
> > > Q5. Also, more recent works of graph self-supervised learning can be considered and cited. For example, Han X, Huang Z, An B, et al. Adaptive Transfer Learning on Graph Neural Networks.
> > >
> > > A5. Thank you for the suggestion. This work also adaptively combines different SSL tasks but the combination happens at the fine-tuning stage and thus requires label information. By contrast, our AutoSSL is proposed to tackle the challenging unsupervised task combination. We have cited and discussed it in the related work section.
> > >
> > > ----
> > > In light of these responses, we hope we have addressed your concerns.  If we have left any notable points of concern unaddressed, please do share and we will attend to these points.

---

### Official Review · Reviewer_3wSZ · 2021-11-03

**Correctness:** 3
**Technical Novelty And Significance:** 3
**Empirical Novelty And Significance:** 3
**Recommendation:** 6
**Confidence:** 4

**Main Review:**

Strengths:
1. Paper is well written and easy to follow.
2. Pseudo-homophily is shown to be a good surrogate measure for mutual information, and combining with search strategies proves effective on several benchmark datasets and tasks.


Weaknesses:
1. Method is somewhat simple and intuitive
2. Method underperforms significantly on clustering task for Photos dataset.

**Summary Of The Paper:**

The proposed work looks at the task of automated self-supervised learning (SSL) on graphs, by using pseudo-homophily as a surrogate objective combined with a search strategy for the proposed approach, AutoSSL .

Homophily is defined as the average of sameness of labels over pairs of connected vertices. Pseudo-homophily is computed by assigning labels based on k-means clustering. The work theoretically shows that maximizing pseudo-homophily is shown to maximize the upper bound of mutual information between pseudo-labels and downstream labels.

As cluster assignments are not differentiable and need search over a large space, the work looks at using evolutionary strategies (AutoSSL-ES) and differentiable search through soft cluster assignments using Gaussian mixture model (AutoSSL-DS).

Experiments are performed on 5 SSL tasks and 8 datasets. AutoSSL based approaches outperform several other unsupervised baselines and perform comparably to supervised baselines when measured using normalized mutual information, accuracy of node classification and psuedo-homophily.

**Summary Of The Review:**

The method is well motivated and quite intuitive, the paper is easy to follow. Extensive experimental results are provided on several tasks and datasets, but statistical significance of improvements is missing, which is needed given large confidence intervals reported.

---

> ### Author Response · Authors · 2021-11-19
> **Response to Reviewer 3wSZ - Part 1**
>
> We appreciate the reviewer's perception of our contributions and thank the reviewer for the insightful questions. Our detailed responses are below.
>
> ----
> Q1. Method is somewhat simple and intuitive
>
> A1. To enable automated self-supervised learning on graphs, we face significant challenges that call for novel solutions. Below we highlight several key challenges and innovations in our work:
>
> (1) **The challenge of the unsupervised setting.** The unsupervised setting is highly challenging in that we do not have an easy-to-get objective to guide the search process, e.g., the validation loss as in traditional NAS papers. To address this issue, we take advantage of an essential property of many real-world graphs, i.e. homophily, to guide the search process. This property is *exclusive for graphs* and most works on NAS do not aim to solve the *unsupervised* AutoML problem.
>
> (2) **The justification of the proposed pseudo-homophily.** To address the aforementioned challenge, we proposed to use pseudo-homophily as the search guidance. We show that increasing pseudo-homophily can indeed help learn better node representations from both theoretical and empirical perspectives.
>
> (3) **Optimization strategy.** First, the pseudo-homophily involves a hard assignment process during clustering, which poses challenges in updating task weights with gradient descent. To address this issue, we convert the hard clustering into soft clustering and optimize the homophily loss proposed in Eq (5).  Second, we cannot use simple gradient descent to update the task weights as homophily loss does not directly involve the task weights. This also makes our work different from traditional NAS works as their outer objective (e.g., validation loss) directly involves the model architecture which is the search goal.
>
> (4) **Necessity of Automated SSL.** Without the knowledge of downstream tasks, we do not know in reality which SSL tasks are most suited for transfer.  Our empirical results demonstrate that different SSL tasks can have distinct impact on different datasets and different tasks. Thus, it is important to take advantage of different SSL tasks while automatically controlling their contributions in a principled way.

---

> > ### Author Response · Authors · 2021-11-19
> > **Response to Reviewer 3wSZ - Part 2**
> >
> >
> > ----
> > Q2. Confidence intervals are reported, but not shown to be statistically significant.
> >
> > A2.  Thanks for your suggestion -- we have included the significance results in Tables 1-3 from paired t-tests to justify the improvement from our framework:
> > * **Significance test.** We used the blue and red numbers to indicate the statistically significant improvements over the best individual task, via paired t-tests with significance levels  of 0.05 and 0.1, respectively. Specifically, p-value indicates the probability that the significance of the results occurred by chance.  As we can see from the tables, the improvements from AutoSSL *over the best baselines on each task* are statistically significant at level 0.05 in most cases.
> >
> > Furthermore, as our goal is to learn generalizable node representations, we also clarify that the combined SSL can achieve good performance *on various downstream tasks as well as different datasets*, while existing methods fail to achieve this goal:
> >
> > * **Maintaining good performance on different downstream tasks.** Greatly improving one of the downstream tasks (clustering or classification) while maintaining the performance on the other task can be considered a significant improvement. Let’s take a look at the example of CoraFull. In Table 2, on node classification AutoSSL vs. MVGRL is 61.10±0.68 vs. 60.56±0.33 (=mild improvement), while on node clustering AutoSSL vs. MVGRL is 0.506±0.01 vs. 0.400±0.01 (=more than 25% relative improvement). Similar significant improvements from AutoSSL can be found in other cases in Table 1, 2 and 3.
> > * **Average rank over different datasets.** AutoSSL is also able to achieve the best performance across different datasets. To see it clearly, we now compute the average ranks of all the methods over eight datasets on clustering and classification and report them in the following two tables (we have also included them in the main content of our paper). We can see that the two AutoSSL variants consistently achieve the best ranks on both clustering and classification.
> >
> > |                |     |     |         |         |     | AutoSSL | AutoSSL    |
> > |----------------|-----|-----|---------|---------|-----|:-------:|:---:|
> > |                | Clu | Par | PairSim | PairDis | DGI | ES      | DS  |
> > | Clustering     | 6.3 | 3.5 | 4.1     | 6.5     | 4.6 | 1.3     | 1.5 |
> > | Classification | 6.8 | 3.9 | 4.9     | 5.4     | 3.9 | 1.8     | 1.4 |
> >
> > |                |       |         |     |      |       |       | AutoSSL | AutoSSL |
> > |----------------|-------|---------|-----|------|-------|:-----:|:-------:|---------|
> > |                | Init. | RawFeat | GAE | VGAE | ARVGA | MVGRL | ES      | DS      |
> > | Clustering     | 6.4   | 7.1     | 4.3 | 4.0  | 6.4   | 4.3   | 1.6     | 2.1     |
> > | Classification | 6.0   | 7.0     | 4.1 | 4.6  | 7.1   | 3.1   | 1.9     | 1.5     |
> >
> > ----
> > Q3. Method underperforms on clustering tasks for Photos dataset.
> >
> > A3. We agree that AutoSSL underperforms GAE on clustering for Photo dataset. However, we note that **GAE only performs well in this case and significantly underperforms on other datasets** (e.g., AutoSSL vs. GAE is 0.726 v.s. 0.545 on Physics for clustering) and classification task (e.g., AutoSSL vs. GAE is on 69.13 vs. 52.57 on ogbn-arxiv for classification). It detracts from our goal of learning generalizable node representations that can *benefit different downstream tasks and datasets*. From the tables we posted in the second answer (A2), we can observe that the AutoSSL variants perform the best on various datasets and different tasks: ES achieve average ranks of 1.6 and 1.9 on clustering and classification; DS achieve average ranks of 2.1 and 1.5 on clustering and classification.
> >
> > -----
> >
> > In light of these responses, we hope we have addressed your concerns, and hope you will consider increasing your score.  If we have left any notable points of concern unaddressed, please do share and we will attend to these points.

---

> > > ### Comment · Reviewer_3wSZ · 2021-12-02
> > > **Reviewer response**
> > >
> > > Thanks for addressing the concerns. I will update my review to reflect this.

---

### Author Response · Authors · 2021-11-29
**Summary of the major revision**

We thank the reviewers for the thorough and detailed reviews on our submission. We summarize major changes that we have made below. All changes are marked in blue in the updated submission.

* We included the statistical significance results in Tables 1-3 (Page 7 and 8) from paired t-tests to justify the improvement from our framework.
* We added the average ranks of all methods in Table 1-3 (Page 7 and 8) and further showed the significance of our work:
  * the proposed AutoSSL achieves **the best average ranks** over *8 datasets* on both classification and clustering tasks.
  * the baselines or individual tasks **fail to maintain good performances** on different downstream tasks and datasets.
* We included the discussion of one recent work [1] on graph self-supervised learning (Section 2, Page 3).

---
[1] Han X, Huang Z, An B, et al. Adaptive Transfer Learning on Graph Neural Networks. KDD 2021

---

### Decision · Program_Chairs · 2022-01-20

**Decision:**

Accept (Poster)

**Comment:**

Strengths:
* Strong empirical study across multiple datasets. However, the gains are not as impressive as for other pretraining domains, such as text or images.
* Interesting formulation of pseudo-homophily as an objective to optimize in the self-supervision stage
* Well-written paper

Weaknesses:
* Novelty may be limited by the fact that the method is essentially learning (or searching) for a weighted average of self-supervised training objectives
* In that case, while the pseudo-homophily angle is interesting, there may be other appropriate baselines for yielding this weighted combination of tasks that are not explored
* There is concern about the degree of empirical improvements on certain datasets